# Critical role of WNK1 in MYC-dependent early mouse thymocyte development

Robert Köchl[1,2]*, Lesley Vanes[1], Miriam Llorian Sopena[1], Probir Chakravarty[1], Harald Hartweger[1], Kathryn Fountain[1], Andrea White[3], Jennifer Cowan[3], Graham Anderson[3], Victor LJ Tybulewicz[1,4]*

[1]The Francis Crick Institute, London, United Kingdom; [2]Kings College London, London, United Kingdom; [3]University of Birmingham, Birmingham, United Kingdom; [4]Imperial College, London, United Kingdom

**Abstract** WNK1, a kinase that controls kidney salt homeostasis, also regulates adhesion and migration in CD4+ T cells. *Wnk1* is highly expressed in thymocytes, and since migration is important for thymocyte maturation, we investigated a role for WNK1 in mouse thymocyte development. We find that WNK1 is required for the transition of double negative (DN) thymocytes through the β-selection checkpoint and subsequent proliferation and differentiation into double positive (DP) thymocytes. Furthermore, we show that WNK1 negatively regulates LFA1-mediated adhesion and positively regulates CXCL12-induced migration in DN thymocytes. Despite this, migration defects of WNK1-deficient thymocytes do not account for the developmental arrest. Instead, we show that in DN thymocytes WNK1 transduces pre-TCR signals via OXSR1 and STK39 kinases, and the SLC12A2 ion co-transporter that are required for post-transcriptional upregulation of MYC and subsequent proliferation and differentiation into DP thymocytes. Thus, a pathway regulating ion homeostasis is a critical regulator of thymocyte development.

**\*For correspondence:**
Robert.Koechl@crick.ac.uk (RK);
Victor.T@crick.ac.uk (VLJT)

**Competing interests:** The authors declare that no competing interests exist.

## Introduction

With No Lysine kinase 1 (WNK1) is a member of a family of four mammalian serine/threonine-specific protein kinases that are broadly conserved across evolution and are found in all multicellular organisms (*Veríssimo and Jordan, 2001*). Mutations in *WNK1* and *WNK4* result in familial hypertension due to altered salt reabsorption in the kidney, because they regulate ion transport in kidney epithelial cells (*Wilson et al., 2001*). WNK kinases phosphorylate and activate the related OXSR1 and STK39 kinases, which in turn phosphorylate and activate the Na+K+Cl- co-transporters SLC12A1 and SLC12A2 and the Na+Cl- co-transporter SLC12A3 (*Rafiqi et al., 2010*; *Thastrup et al., 2012*), allowing Na+, K+, and Cl- ions to enter the cell. Furthermore, they phosphorylate and inhibit the K+Cl- co-transporters SLC12A4, SLC12A5, SLC12A6, SLC12A7 (*Mercado et al., 2016*), blocking K+ and Cl- from leaving the cell. Thus, the net effect of WNK kinase signaling is to promote movement of Na+, K+, and Cl- ions into the cell. Beyond its role in ion homeostasis, WNK1 has been proposed to regulate vesicular trafficking, proliferation and cell volume (*de Los Heros et al., 2018*; *McCormick and Ellison, 2011*).

Unexpectedly, we recently showed that signaling from both the T-cell antigen receptor (TCR) and from the CCR7 chemokine receptor in CD4+ T cells lead to activation of WNK1 (*Köchl et al., 2016*). Furthermore, we found that WNK1 is a negative regulator of TCR- or CCR7-induced adhesion to ICAM1 mediated by LFA1. Conversely, WNK1 is a positive regulator of chemokine-induced migration through OXSR1, STK39, and SLC12A2. As a result, WNK1-deficient T cells home less efficiently to lymphoid organs and migrate more slowly through them. Thus, a pathway that regulates salt homeostasis in the kidney, also controls T-cell adhesion and migration.

*Wnk1* expression levels are particularly high in the thymus (*Shekarabi et al., 2013*), where αβ and γδ T cells develop. Generation of αβ T cells occurs through a series of well-defined developmental subsets. The most immature double negative (DN) thymocytes, expressing neither CD4 nor CD8, can be subdivided into DN1 (CD25⁻CD44⁺CD117⁺, early thymic progenitors, ETP), DN2 (CD25⁺CD44⁺CD117⁺), DN3 (CD25⁺CD44⁻CD117⁻) and DN4 (CD25⁻CD44⁻CD117⁻) subsets (*Bhandoola et al., 2007*; *Godfrey et al., 1993*; *Yui et al., 2010*). Subsequently, the cells upregulate CD8 and then CD4, becoming CD4⁻CD8⁺immature single positive (ISP) cells and then CD4⁺CD8⁺double positive (DP) thymocytes. Finally, they lose expression of either CD4 or CD8 to become CD4⁺ or CD8⁺ single positive (4SP or 8SP) cells and emigrate from the thymus as CD4⁺ or CD8⁺ T cells.

To generate T cells that have successfully rearranged both TCRα and TCRβ genes and express an αβTCR that is both restricted by self-MHC and self-tolerant, thymocytes need to pass three checkpoints (*Carpenter and Bosselut, 2010*). Thymic progenitors migrate from the bone marrow via the blood, entering the thymus at the cortico-medullary junction as DN1 cells (ETP). DN2 and DN3 thymocytes begin to re-arrange TCRβ genes and migrate to the sub-capsular zone of the cortex. If rearrangement is successful, TCRβ protein binds to pre-Tα and together with CD3γ, CD3δ, CD3ε and CD3ζ forms the pre-TCR (*Yamasaki and Saito, 2007*). Signals from the pre-TCR within DN3 cells result in an increase in cell size and expression of CD27 andCD28, changes which can be used to distinguish DN3a and DN3b cells before and after pre-TCR signaling respectively (*Hoffman et al., 1996*; *Taghon et al., 2006*; *Teague et al., 2010*; *Williams et al., 2005*). Pre-TCR signals are required for the survival and proliferative expansion of DN3b and DN4 thymocytes, and subsequent differentiation into ISP and DP cells, a checkpoint termed β-selection (*Kreslavsky et al., 2012*; *Mingueneau et al., 2013*). Progression through this checkpoint is also supported by signals from CXCR4 and NOTCH1 (*Janas et al., 2010*; *Maillard et al., 2006*; *Trampont et al., 2010*).

Subsequently, DP thymocytes, now in the cortex, rearrange TCRα genes. If they successfully make an αβTCR that can bind weakly to self-peptide-MHC complexes, the cells undergo positive selection into 4SP and 8SP cells and migrate into the medulla (*Klein et al., 2014*). Alternatively, if the αβTCR binds strongly to self-peptide-MHC, auto-reactive T cells are eliminated by negative selection.

In view of the important role of migration during thymocyte development, the high expression of *Wnk1* in the thymus, and our previous work showing that WNK1 regulates adhesion and migration of CD4⁺ T cells, we investigated a potential role for WNK1 during thymocyte development. We find that WNK1 is required for DN thymocytes to progress past the β-selection checkpoint. We show that in DN thymocytes, WNK1 is a negative regulator of LFA1-mediated adhesion and a positive regulator of chemokine-induced migration. However, changes in migration are unlikely to cause the developmental arrest in WNK1-deficient thymocytes. Instead, WNK1, acting through OXSR1, STK39 and SLC12A2, is required for pre-TCR-induced post-transcriptional upregulation of MYC, and subsequent proliferation of DN4 thymocytes and differentiation into DP cells. Thus, a pathway that regulates ion homeostasis is a critical and previously unknown regulator of early thymocyte development.

## Results

### WNK1 pathway genes expressed throughout thymic development

To extend the previous observation that *Wnk1* is expressed at a high level in the thymus (*Shekarabi et al., 2013*), we re-analyzed RNAseq data from multiple stages of thymic development (*Hu et al., 2013*), focusing on expression of WNK-family kinases and proteins in the WNK pathway. We found that *Wnk1* is the only member of the WNK-family expressed in thymocytes, with high expression at all stages of development (*Figure 1—figure supplement 1*). Substantially more *Oxsr1* than *Stk39* is expressed at all stages, except for 4SP thymocytes where the expression of *Stk39* exceeds levels of *Oxsr1*. Of the Na⁺K⁺Cl⁻ and Na⁺Cl⁻ co-transporters, only *Slc12a2* is expressed in the thymus. Finally, there is substantial expression of the *Slc12a6* and *Slc12a7* K⁺Cl⁻ co-transporters. Thus, multiple genes in the WNK1 pathway are expressed throughout thymic development.

## Essential role for WNK1 in thymocyte development

To test whether WNK1 is required during T-cell development, we used the CD2-Cre transgene to delete loxP-flanked (floxed) alleles of *Wnk1* (*Wnk1*[fl]) starting at the DN1 (ETP) stage of development (*de Boer et al., 2003*; *Siegemund et al., 2015*). We compared *Wnk1*[fl/fl]CD2-Cre mice with control *Wnk1*[fl/+]CD2-Cre mice in which thymocytes have homozygous or heterozygous loss of *Wnk1* respectively. We found that loss of WNK1 resulted in a 10-fold reduction in total thymocyte numbers and a progressive loss of cells from the DN4 stage onwards, with a 92% reduction in numbers of ISP thymocytes and a 98–99% reduction in DP, 4SP, and 8SP thymocytes (*Figure 1A–C*). In contrast, there was no change in the numbers of TCRγδ[+] thymocytes. Analysis of *Wnk1* mRNA levels showed that while *Wnk1* expression was lost in DN and ISP populations in thymi from *Wnk1*[fl/fl]CD2-Cre mice, from the DP stage onwards the expression levels were similar to or higher in comparison to control animals (*Figure 1D*), demonstrating that the few remaining DP cells are those that have failed to delete *Wnk1*. Hence, WNK1 is absolutely required for the development of DP thymocytes.

Next, we evaluated whether the kinase activity of WNK1 is required for thymic development. We generated compound heterozygous mice with both a kinase-inactive allele (*Wnk1*[D368A]) and a floxed allele of *Wnk1* and a tamoxifen-inducible Cre in the *ROSA26* locus (*ROSA26*[CreERT2], RCE) and generated bone marrow chimeras by reconstituting irradiated RAG1-deficient mice with marrow from either *Wnk1*[fl/D368A]RCE mice or *Wnk1*[fl/+]RCE mice as a control. Following tamoxifen treatment of these chimeras, lymphocytes are the only cell type in which all cells delete the floxed *Wnk1* allele, leaving them with either a wild-type or D368A *Wnk1* allele. We found that after 7d of tamoxifen treatment, development of thymocytes expressing only WNK1-D368A was again very strongly blocked from the DN4 stage onwards (*Figure 1E,F*), demonstrating that WNK1 kinase activity is required for thymocyte development from the DN to DP stages.

## WNK1 is required for thymocytes to develop past the pre-TCR checkpoint

The block in T-cell development between the DN3 and DN4 and subsequent DP stages in the absence of WNK1 coincides with the β-selection checkpoint, where DN3 cells that have productively rearranged the TCRβ locus and express TCRβ protein assemble the pre-TCR. Subsequent pre-TCR signals result in survival and proliferative expansion of DN4 cells, and differentiation into DP cells. To determine if this β-selection checkpoint might be compromised in WNK1-deficient thymocytes, we used cell size and expression of CD28 to separate DN3 thymocytes into DN3a (small, CD28[-]) and DN3b (large, CD28[+]) cells (*Hoffman et al., 1996*; *Teague et al., 2010*; *Williams et al., 2005*), which correspond to cells before and after the checkpoint respectively. Loss of WNK1 results in no change in the number of DN3a cells, but a substantial drop in DN3b and DN4 cells, consistent with a block at the pre-TCR β-selection checkpoint (*Figure 2A*). To evaluate if this block could be due to a failure to assemble the pre-TCR, we assessed expression of pre-TCR components. Flow cytometric analysis of DN2 and DN3 thymocytes showed that a similar fraction (~10%) of both control and WNK1-deficient cells expressed intracellular TCRβ, corresponding to cells that have successfully rearranged the TCRβ locus, indicating that loss of WNK1 does not impair TCRβ locus rearrangement (*Figure 2B*). A similar analysis of DN4 cells showed that most control cells expressed intracellular TCRβ, as would be expected for cells successfully undergoing β-selection. In contrast, only ~10% of WNK1-deficient DN4 cells express intracellular TCRβ, similar to the percentage in DN3 cells (*Figure 2B*). Thus, in the absence of WNK1, there is no selective expansion of TCRβ[+] cells. Cell-surface levels of CD3ε and pre-Tα in WNK1-deficient DN2 and DN3 cells were comparable to controls, while CD3ε levels were increased in WNK1-deficient DN4 cells (*Figure 2C*). Taken together, these results show that the developmental block in WNK1-deficient DN thymocytes is unlikely to be due to a failure to make pre-TCR subunits. Instead, the phenotype may be caused by failure of pre-TCR signaling.

To directly test this, we made use of an approach in which anti-CD3ε antibodies are injected into RAG1-deficient mice that have arrested thymocyte development at the DN3 stage because of a failure to rearrange TCR genes. RAG1-deficient DN3 thymocytes express small amounts of the CD3 proteins on the cell surface and binding of anti-CD3ε mimics a pre-TCR signal, resulting in differentiation of the thymocytes into DP cells (*Levelt et al., 1993*; *Shinkai and Alt, 1994*). As expected, injection of anti-CD3ε into control *Wnk1*[fl/+]*Rag1*[-/-]CD2-Cre mice resulted in the generation of large numbers of DP thymocytes. By stark contrast, DP thymocytes were almost completely absent in anti-

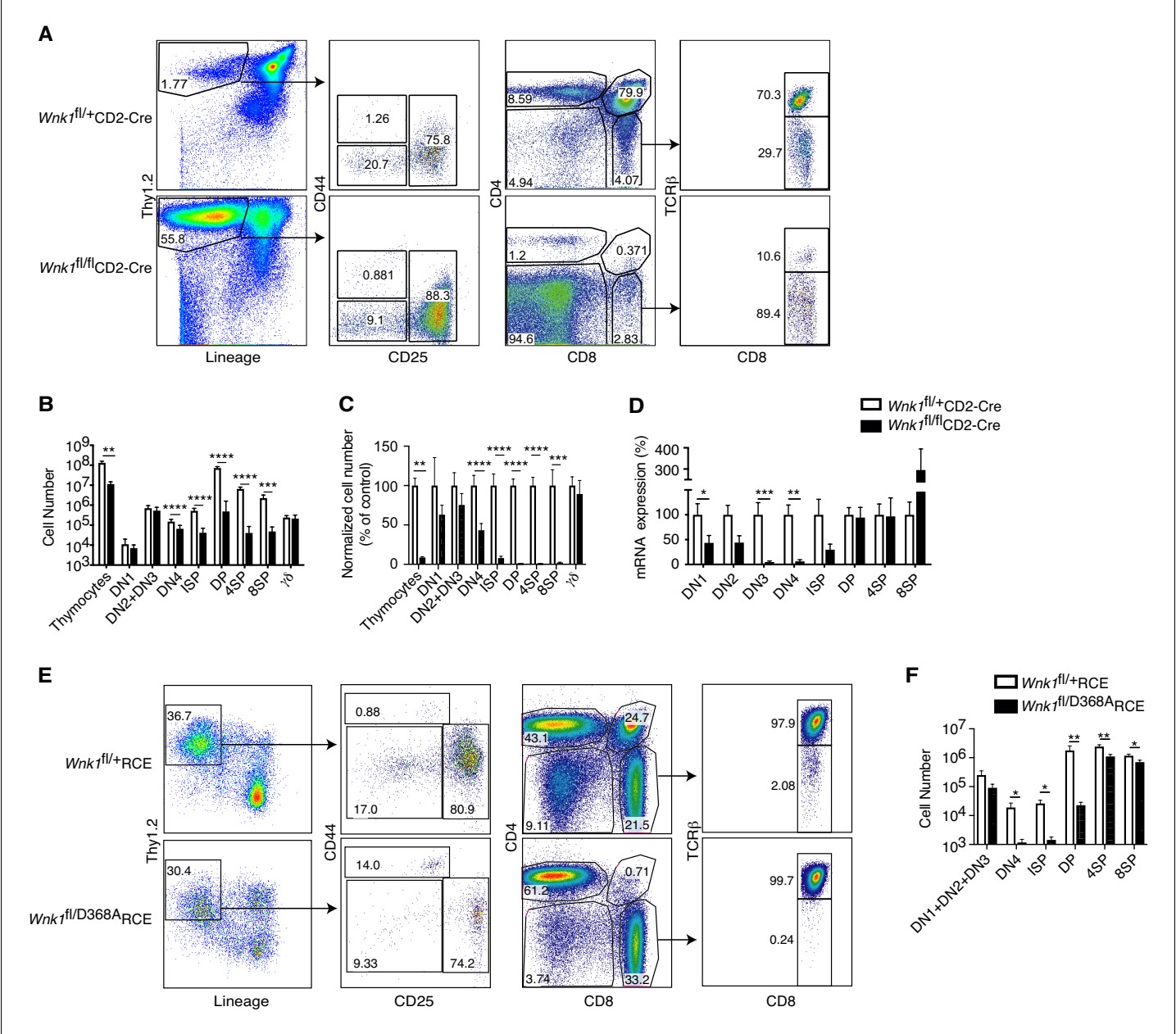

**Figure 1.** Essential role for WNK1 in thymocyte development. (A) Flow cytometric analysis of thymocytes from $Wnk1^{fl/+}$CD2-Cre and $Wnk1^{fl/fl}$CD2-Cre mice showing double negative (DN, Thy1.2$^+$Lineage$^-$) thymocytes subdivided into DN1 (CD44$^+$CD25$^-$), DN2+DN3 (CD25$^+$), and DN4 (CD44$^-$CD25$^-$) subsets, immature single positive (ISP, CD4$^-$CD8$^+$TCRβ$^-$), double positive (DP, CD4$^+$CD8$^+$), CD4$^+$ single positive (4SP, CD4$^+$CD8$^-$), and CD8$^+$ single positive (8SP, CD4$^-$CD8$^+$TCRβ$^+$) thymocytes. Numbers show % of cells falling into each gate. (B) Mean ± SEM number of total thymocytes and of thymocytes at each developmental stage defined using the gates in A, as well as γδ thymocytes (CD4$^-$TCRγ$^+$). (C) Mean ± SEM number of cells in each thymocyte population in $Wnk1^{fl/fl}$CD2-Cre animals normalized to $Wnk1^{fl/+}$CD2-Cre controls (set to 100%). (D) Mean ± SEM $Wnk1$ mRNA levels in thymic subsets from $Wnk1^{fl/+}$CD2-Cre and $Wnk1^{fl/fl}$CD2-Cre mice, normalized to $Wnk1^{fl/+}$CD2-Cre controls (set to 100%) measured by Q-PCR from exon 1 to exon 2. CD44 levels were used to separate DN2 (CD44$^+$) and DN3 (CD44$^-$) thymocytes. (E) Flow cytometric analysis of $Rag1^{-/-}$ radiation chimeras reconstituted with bone marrow from $Wnk1^{fl/+}$RCE and $Wnk1^{fl/D368A}$RCE mice, treated with tamoxifen and analyzed 7 d later showing gating for thymocyte subsets as in A, with DN (Thy1.2$^+$Lineage$^-$) thymocytes pre-gated on CD4$^-$CD8$^-$ cells. (F) Mean ± SEM number of cells in thymocyte subsets of $Rag1^{-/-}$ chimeras as in E. *0.01 < p < 0.05, ** 0.001 < p < 0.01, and ***p<0.001. Significance calculated by Mann-Whitney test. Sample sizes: five $Wnk1^{fl/+}$CD2-Cre and six $Wnk1^{fl/fl}$CD2-Cre mice (B, C), seven (D), and six (F) of each genotype. Data are pooled from two independent experiments. The online version of this article includes the following figure supplement(s) for figure 1:

**Figure supplement 1.** Expression of genes encoding WNK pathway proteins.

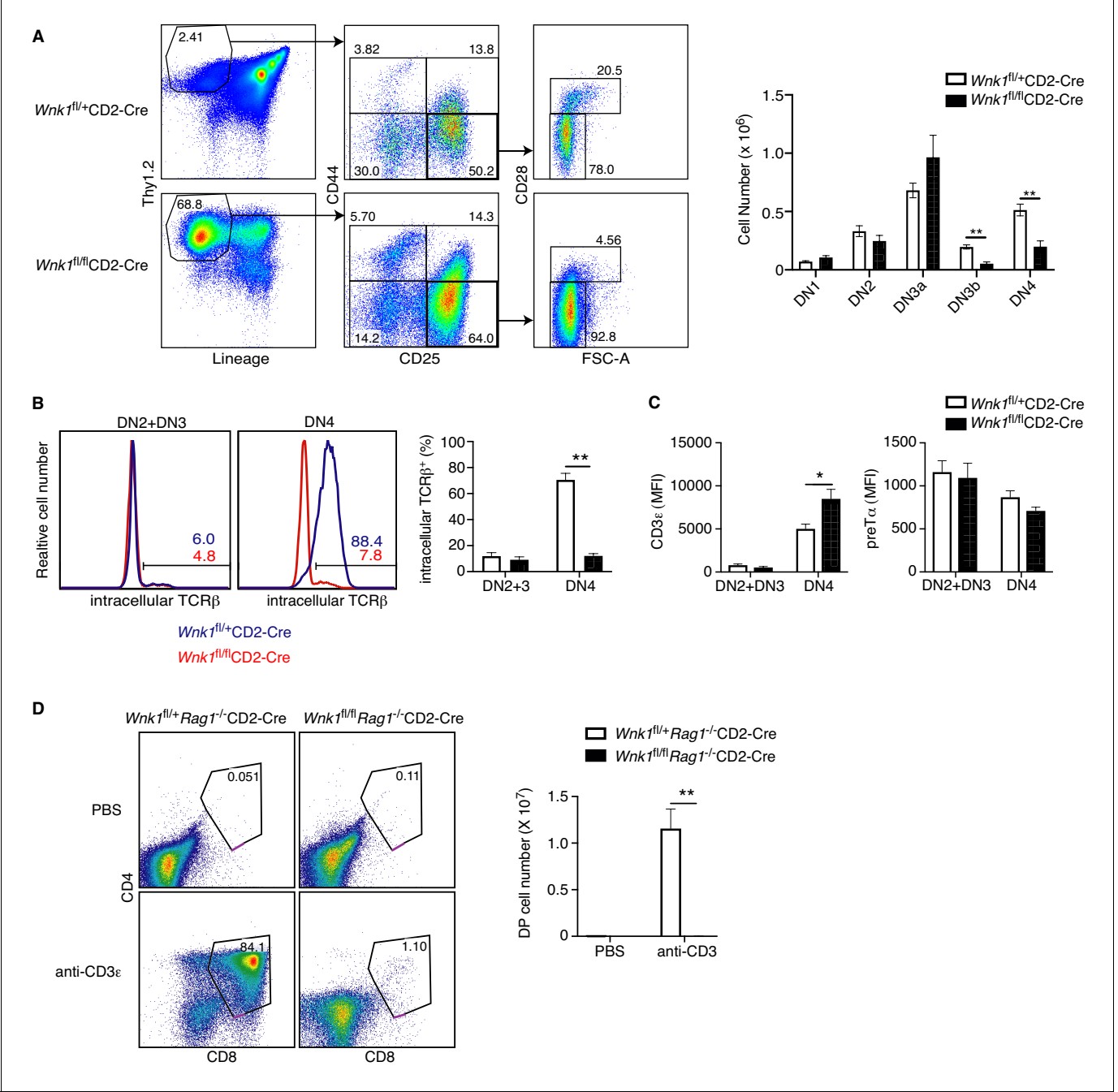

**Figure 2.** WNK1 is required for thymocytes to develop past the pre-TCR checkpoint. (A) Left: flow cytometric analysis of Thy1.2[+]Lineage[-] thymocytes separated by expression of CD44 and CD25, and then DN3 (CD44[-]CD25[+]) cells separated into DN3a (small, FSC[low]CD28[-]) and DN3b (large, FSC[high]CD28[+]) cells. Right: Mean ± SEM cell numbers in DN subsets. (B) Left: Flow cytometric analysis of thymocytes from *Wnk1*[fl/+]CD2-Cre and *Wnk1*[fl/fl]CD2-Cre mice showing intracellular TCRβ levels in DN2+DN3 and DN4 subsets. Numbers indicate the percentage of TCRβ[+] cells. Right: Mean ± SEM percentage of DN2+DN3 and DN4 cells that are intracellular TCRβ[+]. (C) Mean ± SEM of CD3ε or pre-Tα surface levels measured as mean fluorescence intensity (MFI) in flow cytometry of DN2+DN3 and DN4 thymocytes from *Wnk1*[fl/+]CD2-Cre and *Wnk1*[fl/fl]CD2-Cre mice. (D) Left: flow cytometric analysis of CD4 and CD8 levels on thymocytes from *Wnk1*[fl/+]*Rag1*[-/-]CD2-Cre and *Wnk1*[fl/fl]*Rag1*[-/-]CD2-Cre mice harvested 4 d after intraperitoneal injection with PBS or anti-CD3ε antibody. Gate indicates DP thymocytes and number shows percentage of cells in the gate. Right: mean ± SEM number of DP thymocytes defined using the gates on the left. *0.01<p<0.05, and **p<0.01. Significance calculated by Mann-Whitney test. Sample sizes: six of each genotype (A, B), eight *Wnk1*[fl/+]CD2-Cre and ten *Wnk1*[fl/fl]CD2-Cre mice (C), four *Wnk1*[fl/+]*Rag1*[-/-]CD2-Cre and *Wnk1*[fl/fl]*Rag1*[-/-]CD2-Cre mice injected with PBS and five *Wnk1*[fl/+]*Rag1*[-/-]CD2-Cre and *Wnk1*[fl/fl]*Rag1*[-/-]CD2-Cre mice injected with anti-CD3ε (D). Data are pooled from two independent experiments.

CD3ε treated *Wnk1*<sup>fl/fl</sup>*Rag1*<sup>-/-</sup>CD2-Cre mice (*Figure 2D*), indicating that WNK1 is essential for pre-TCR signaling that controls the DN-DP transition in thymocytes.

## RNAseq of WNK1-deficient thymocytes reveals significant changes in five distinct processes

To gain further insight into the role of WNK1 in this developmental transition, we used RNA sequencing (RNAseq) to analyze the transcriptome of DN3 and DN4 cells sorted from the thymi of *Wnk1*<sup>fl/+</sup>*Rag1*<sup>-/-</sup>CD2-Cre and *Wnk1*<sup>fl/fl</sup>*Rag1*<sup>-/-</sup>CD2-Cre mice, harvested 0, 3, 24, and 48 hr after injection with anti-CD3ε (*Figure 3A,B*, *Supplementary file 1*). Principal component analysis (PCA) of the results revealed that the biggest differences between the samples were caused by anti-CD3ε stimulation and not by genotype (*Figure 3C*). Both, WNK1-deficient thymocytes and control cells followed a similar trajectory on the PCA plot as a function of time after anti-CD3ε injection, suggesting that pre-TCR signaling is able to induce similar transcriptional changes in control and WNK1-deficient thymocytes. Nevertheless, comparing the transcriptomes of control and WNK1-deficient thymocytes, we identified a large number of statistically significant differentially expressed genes (DEGs) at each time point, including in unstimulated cells (*Figure 3D*, *Supplementary files 2*, *3*).

To further characterize these changes, we carried out a process enrichment analysis taking into account all DEGs at each time point and found that most of the identified processes fit into five functional groups: genes associated with TCR signaling, protein translation, cytoskeleton, adhesion and cell cycle (*Figure 3E*). The TCR signaling and translation signatures were most enriched at the 0 hr time point, cytoskeleton and adhesion at 48 hr, and cell cycle at 24 hr. Thus, WNK1 deficiency causes changes in several key biochemical pathways, any or all of which could be causing the developmental arrest.

## WNK1 deficiency does not perturb early pre-TCR-induced signaling and protein translation

We investigated the potential contribution of each of the perturbed processes to the developmental block caused by loss of WNK1. Inspection of the TCR signaling signature showed that WNK1-deficient thymocytes had decreased levels of mRNA for *Cd3d*, *Cd3e*, *Cd3g*, *Lck,* and *Syk*, coding for CD3 proteins and the LCK and SYK tyrosine kinases and an increase in *Lcp2* mRNA, which codes for the SLP76 adapter protein (*Figure 4A*), suggesting that pre-TCR signaling may be perturbed, though we note that at least for CD3ε, reduced mRNA level in WNK1-deficient cells did not result in reduced cell-surface protein level (*Figure 2C*). However, analysis of transcriptional changes driven by pre-TCR stimulation showed that both control and WNK1-deficient cells upregulated *Nur77*, *Cd27*, and *Cd28*, and downregulated *Il2ra*, *Hes1*, *Rag2*, and *Ptcra* to a similar extent (*Figure 4B*). Furthermore, anti-CD3ε-induced downregulation of surface levels of CD25 (encoded by *Il2ra*) also occurred normally in WNK1-deficient cells (*Figures 3B* and *4C*). These results imply that loss of WNK1 does not have a major effect on pre-TCR signaling, at least for the first 24 hr after stimulation.

The most highly enriched processes in the RNAseq analysis were associated with protein translation, with many of the genes in these processes being downregulated in WNK1-deficient cells, particularly at the 0 hr time point (*Figure 3E*, *Figure 4—figure supplement 1A*). We hypothesized that this could lead to lower overall protein synthesis and might explain the developmental block. To test this, we measured levels of phosphorylated S6 (pS6), a hallmark of active translation, in DN3 and DN4 thymocytes from mice 24 hr after injection of anti-CD3ε, and also directly tested the capacity of the cells to synthesize proteins using a puromycin incorporation assay. We found that pS6 levels in WNK1-deficient cells were similar to those in control cells before stimulation and higher than in controls 24 hr after anti-CD3ε stimulation (*Figure 4—figure supplement 1B*). Furthermore, rates of protein synthesis were increased to a similar extent by anti-CD3ε in both control and mutant thymocytes (*Figure 4—figure supplement 1C*). Thus, despite the reduced expression of genes associated with translation, the absence of WNK1 does not cause substantial changes in rates of protein synthesis, and these are unlikely to explain the developmental block.

## WNK1 regulates thymocyte migration and adhesion

In the process enrichment analysis of the RNAseq data we identified cytoskeleton and adhesion signatures. At the 24 hr time point WNK1-deficient thymocytes had increased expression of *Itgb2* and

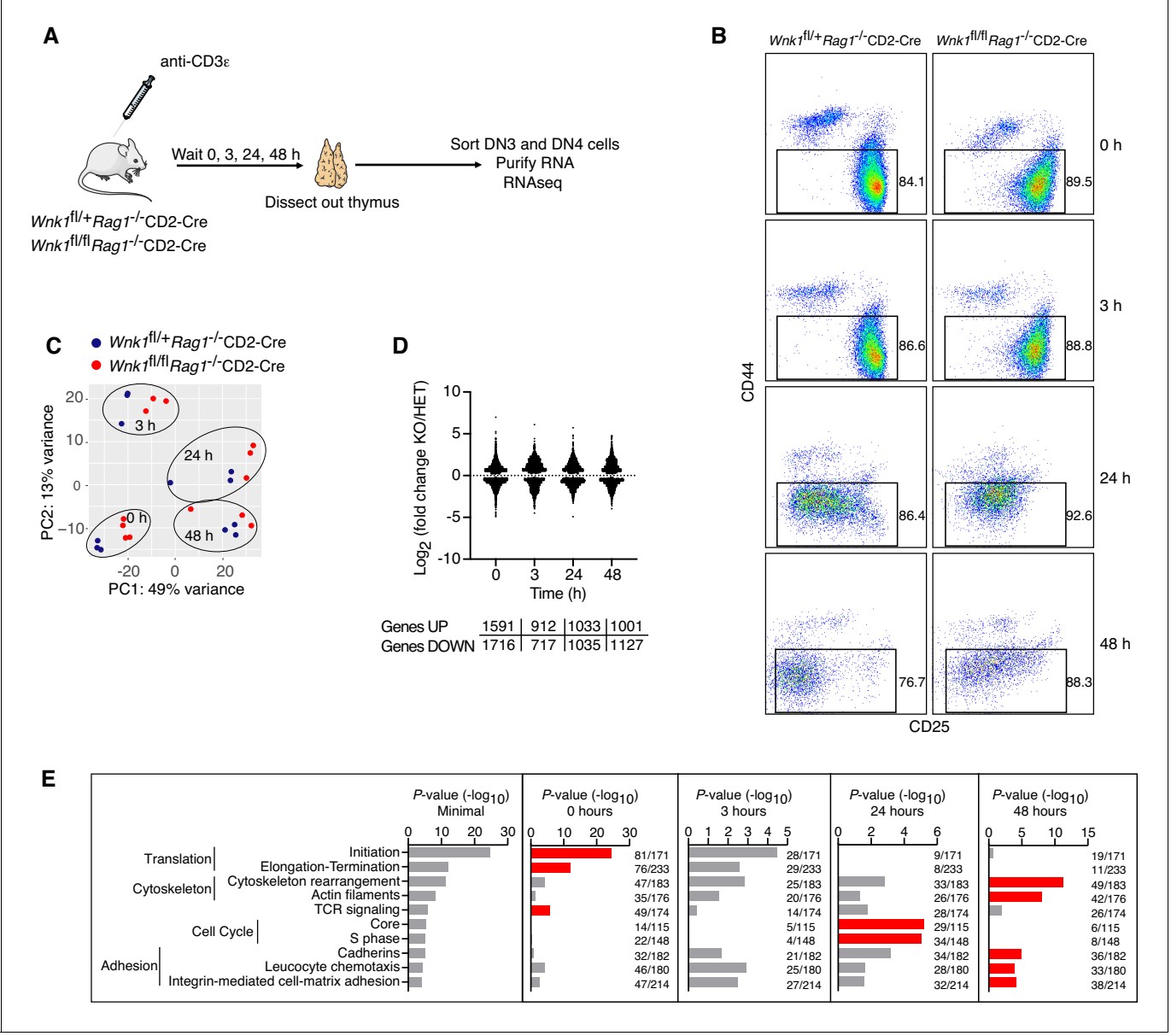

**Figure 3.** RNAseq of WNK1-deficient thymocytes reveals significant changes in five distinct processes. (A) Thymi were harvested from *Wnk1*fl/+*Rag1*-/-CD2-Cre and *Wnk1*fl/fl*Rag1*-/-CD2-Cre mice that were either untreated (0 hr) or had been injected with anti-CD3ε antibody 3, 24 or 48 hr earlier. DN3+DN4 thymocytes were sorted, RNA purified and analyzed by RNAseq, results of which are shown in C-E. (B) Flow cytometric analysis of DN thymocytes from the experiment described in A. Gates indicate cells (Thy1.2+, CD44-) that were sorted for further RNAseq analysis. (C) Principal component analysis using expression of all normalized genes for all samples from the experiment described in A. (D) Log2 fold changes and total number of statistically significant differentially expressed genes (DEGs) in samples from *Wnk1*fl/fl*Rag1*-/-CD2-Cre (KO) mice compared to *Wnk1*fl/+*Rag1*-/-CD2-Cre (HET) mice at each time point (*Supplementary files 2*, *3*). (E) Metacore process analysis of DEGs. Selected processes and their *P*-values are shown at each time point; graph on left shows the lowest *P*-value for each process from any of the four time points. Red bars indicate time point with the lowest *P*-values for each process (Minimal). Ratios indicate the number of DEGs at each time point over the number of all genes in the indicated processes. Significance calculated by Wald's test (D) and Hypergeometric test (E). Sample sizes: three *Wnk1*fl/+*Rag1*-/-CD2-Cre and four *Wnk1*fl/fl*Rag1*-/-CD2-Cre mice for 0 hr time point, three mice of each genotype for all other time points; data are from one experiment.

*Itgal*, the subunits of LFA1, as well as *Rap1a* and *Rap2a* that encode GTPases required for integrin signaling, and several chemokine receptors (*Ccr2*, *Ccr5*, and *Cxcr5*), although *Cxcr4* expression was reduced (*Figure 5A*); some of the same changes were also visible 48 hr after stimulation. In agreement with this, cell-surface LFA1 was increased and CXCR4 was decreased on both WNK1-deficient DN2+DN3 and DN4 thymocytes (*Figure 5B*). We have previously shown that WNK1 is a negative

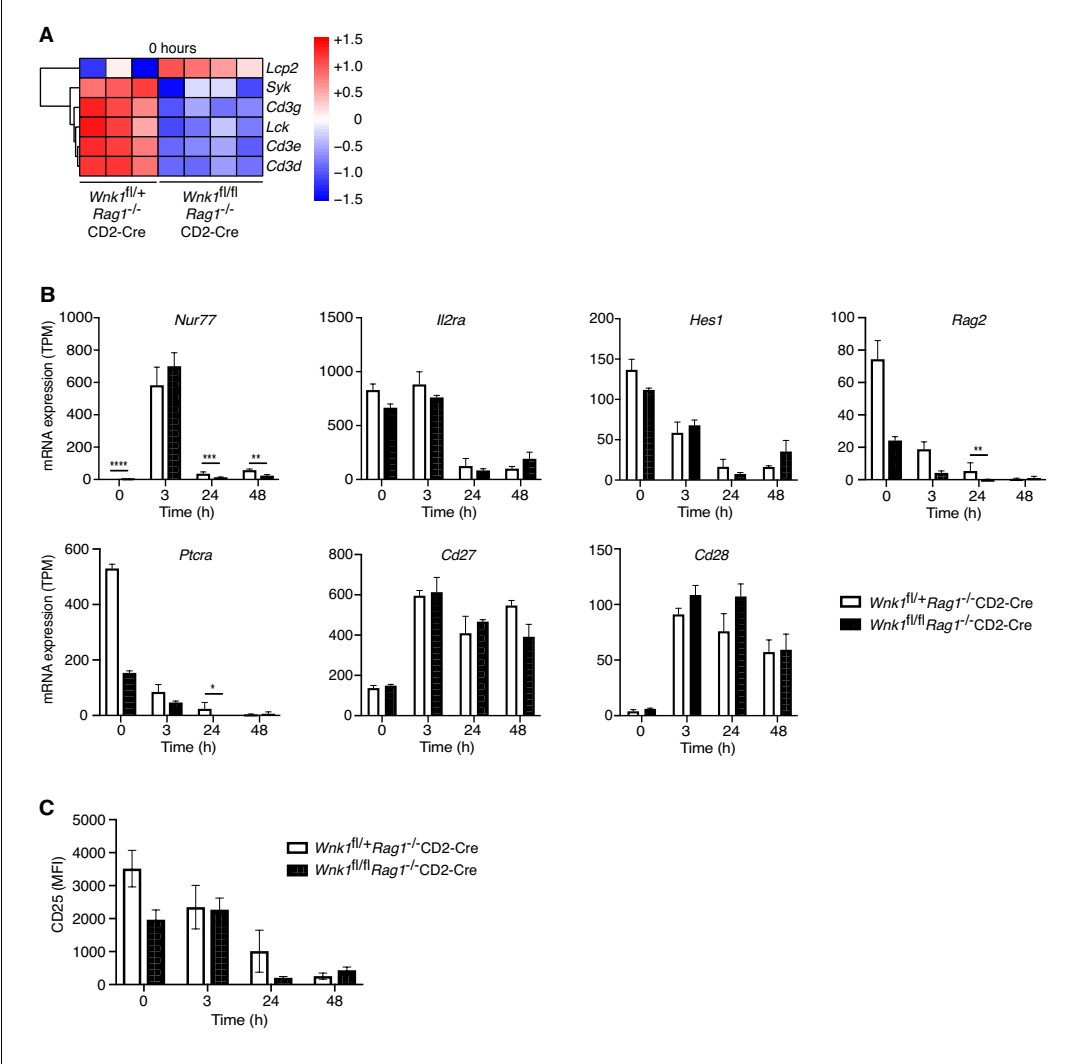

**Figure 4.** WNK1 deficiency does not perturb early pre-TCR-induced signaling. (**A**) Analysis of RNAseq experiment described in *Figure 3A*, showing heatmap and hierarchical clustering of selected genes from the TCR signaling signature at 0 hr in $Wnk1^{fl/+}Rag1^{-/-}$CD2-Cre and $Wnk1^{fl/fl}Rag1^{-/-}$CD2-Cre mice. Red and blue colors indicate increased or decreased expression of indicated genes relative to the mean expression of each row using normalized $\log_2$ expression values. (**B**) Mean ± SEM expression of selected genes; TPM, transcripts per million reads. (**C**) Mean ± SEM surface levels of CD25 on CD44⁻ thymocytes analyzed as in *Figure 3B*. \*$0.01 < p < 0.05$, \*\* $0.001 < p < 0.01$, \*\*\*$0.0001 < p < 0.001$, \*\*\*\*$p<0.0001$. Significance displayed in (**B**) is from the analysis of the RNAseq data and was calculated by the Wald test (*Supplementary file 3*). Sample sizes: three $Wnk1^{fl/+}Rag1^{-/-}$CD2-Cre and four $Wnk1^{fl/fl}Rag1^{-/-}$CD2-Cre mice for 0 hr time point and three mice of each genotype for all other time points. Data are from one experiment.

The online version of this article includes the following figure supplement(s) for figure 4:

**Figure supplement 1.** WNK1 deficiency does not perturb protein translation.

regulator of LFA1-mediated adhesion and a positive regulator of chemokine-induced migration in CD4⁺ T cells (*Köchl et al., 2016*). These functions, together with the transcriptional changes seen here, suggest that WNK1 may also regulate adhesion and migration of DN thymocytes, which could account for the developmental block.

To investigate if CXCR4 can activate the WNK1 pathway in thymocytes, we stimulated primary mouse DN3 cells from $Rag1^{-/-}$ mice with CXCL12, the ligand for CXCR4, in the presence or absence of the WNK1 inhibitor WNK463 and analyzed phosphorylation of OXSR1 on Ser325, a WNK1 substrate (*Vitari et al., 2005*). We found that CXCL12 resulted in the phosphorylation of OXSR1, which was abrogated in WNK1 inhibitor-treated cells (*Figure 5C*). Thus, CXCR4 transduces signals through WNK1 leading to the phosphorylation and, presumably, activation of OXSR1.

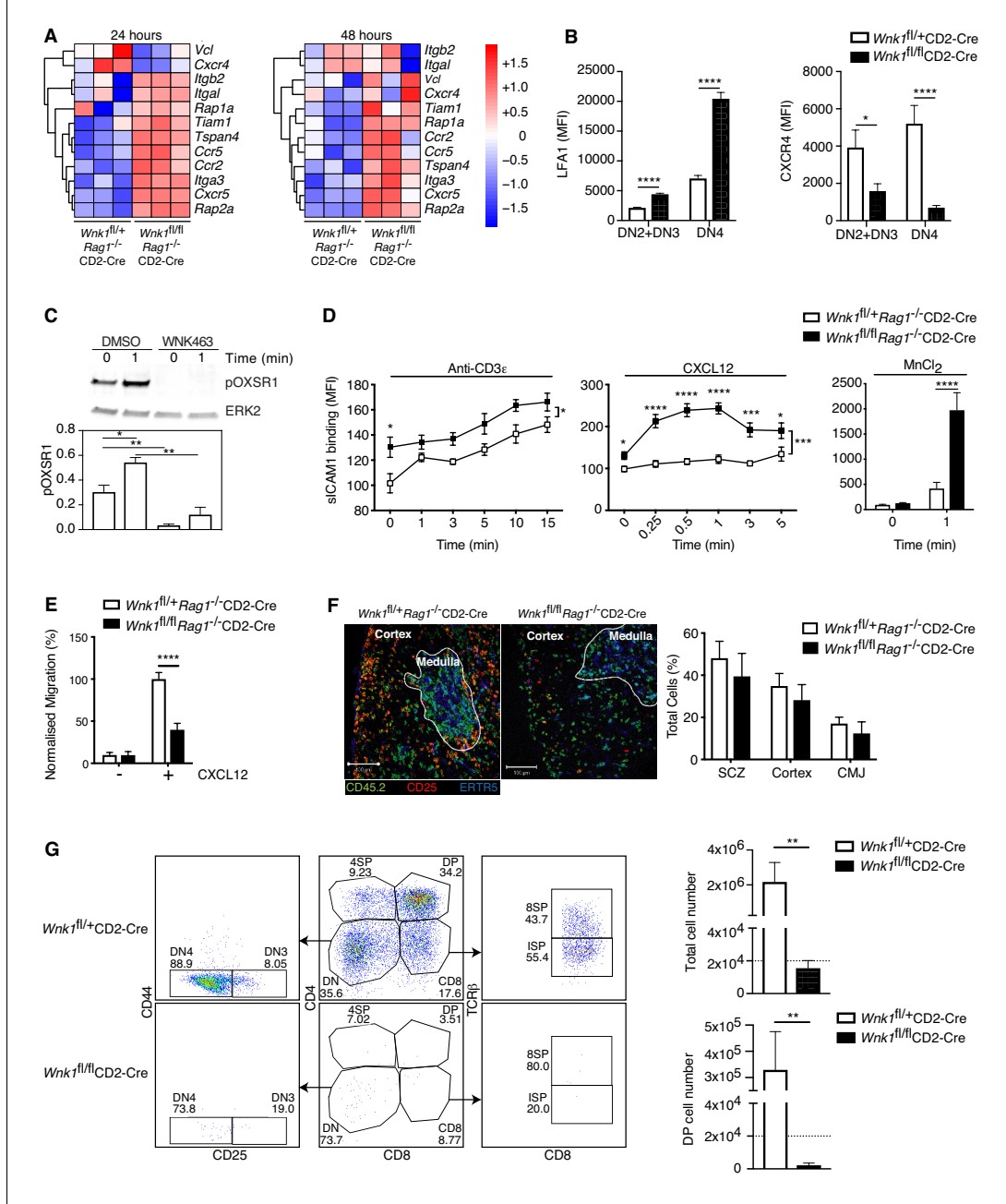

**Figure 5.** WNK1 regulates thymocyte migration and adhesion. (**A**) Heatmap and hierarchical clustering of selected genes from the adhesion signature at 24 and 48 hr after injection of anti-CD3ε into *Wnk1*^fl/+^*Rag1*^-/-^CD2-Cre and *Wnk1*^fl/fl^*Rag1*^-/-^CD2-Cre mice from the experiment described in *Figure 3A*. (**B**) Mean ± SEM LFA1 and CXCR4 surface levels in DN2+DN3 and DN4 thymocytes, pre-gated on double negative (TCRβ⁻, Thy1.2⁺, Lineage⁻) thymocytes, from *Wnk1*^fl/+^CD2-Cre and *Wnk1*^fl/fl^CD2-Cre mice. (**C**) Top, representative immunoblot analysis of phosphorylated OXSR1 (pOXSR1) and ERK2 in total *Rag1*^-/-^ thymocytes incubated with DMSO or WNK463 and stimulated for 0 or 1 min with CXCL12. Below, quantification of pOXSR1 normalized to the abundance of ERK2. (**D**) Binding of soluble ICAM1 complexes to mouse DN2+DN3 thymocytes from *Wnk1*^fl/+^*Rag1*^-/-^CD2-Cre and *Wnk1*^fl/fl^*Rag1*^-/-^CD2-Cre mice in response to treatment for various times with anti-CD3ε, CXCL12 or MnCl₂. (**E**) Migration of mouse DN2+DN3 thymocytes from the top chamber to the bottom chamber of a Transwell plate in the presence (+) or absence (-) of CXCL12. (**F**) Left: representative images of frozen thymus sections from lethally irradiated B6.SJL mice, reconstituted with a 80:20 mixture of bone marrow from B6.SJL (CD45.2⁻) and either *Wnk1*^fl/+^*Rag1*^-/-^CD2-Cre or *Wnk1*^fl/fl^*Rag1*^-/-^CD2-Cre mice (both CD45.2⁺), stained with antibodies to the indicated antigens, as well as antibodies to CD11b and CD11c (not shown). Right: distribution of *Wnk1*^fl/+^*Rag1*^-/-^CD2-Cre or *Wnk1*^fl/fl^*Rag1*^-/-^CD2-Cre derived DN2+DN3 (CD25⁺CD45.2⁺CD11b⁻CD11c⁻) thymocytes between the subcapsular zone (SCZ), cortex and cortico-medullary junction (CMJ) as defined by the ERTR5 staining for medullary epithelial cells. (**G**) Left: flow cytometric analysis of DN2+DN3 thymocytes from *Wnk1*^fl/+^CD2-Cre and *Wnk1*^fl/fl^CD2-Cre mice after culturing on OP9-DL1 cells for 7 d. Gates indicate DN (CD4⁻CD8⁻), DN3 (CD4⁻CD8⁻CD25⁺CD44⁻), DN4 (CD4⁻CD8⁻CD25⁻CD44⁻), ISP (CD4⁻CD8⁺TCRβ⁻)

*Figure 5 continued on next page*

*Figure 5 continued*

DP (CD4$^+$CD8$^+$), 4SP (CD4$^+$CD8$^-$) and 8SP (CD4$^-$CD8$^+$TCRβ$^+$) cells. Right: mean ± SEM number of all cells and number of DP cells recovered after 7 d. Dashed line indicates number of cells seeded into the cultures on day 0. *0.01 < p < 0.05, ** 0.001 < p < 0.01, ***0.0001 < p < 0.001, ****p<0.0001. Significance calculated by Mann-Whitney test. Sample sizes: three of each genotype (A), eight *Wnk1*$^{fl/+}$CD2-Cre and ten *Wnk1*$^{fl/fl}$CD2-Cre mice (B), five (C), five *Wnk1*$^{fl/+}$CD2-Cre and seven *Wnk1*$^{fl/fl}$CD2-Cre mice for the anti-CD3ε- and MnCl$_2$-induced adhesion assays and five and six for the CXCL12-induced adhesion assays (D), six (E, G), and four (F) of each genotype. Data are from one experiment (A) or pooled from two (B, D–G) or five (C) experiments.

These results suggest that, as in CD4$^+$ T cells, WNK1 may also regulate CXCR4-induced adhesion and migration in DN thymocytes, possibilities we tested directly. We found that in response to stimulation through either CD3ε or CXCR4, or with Mn$^{2+}$ which directly activates LFA1, WNK1-deficient DN3 thymocytes showed increased binding of ICAM1, indicating increased LFA1-mediated adhesion (*Figure 5D*). Furthermore, WNK1-deficient DN3 cells had reduced migration in response to CXCL12 (*Figure 5E*). Thus, WNK1 is a negative regulator of LFA1-mediated adhesion and a positive regulator of CXCL12-induced migration in DN3 thymocytes.

Loss of CXCR4 results in defective thymocyte development in part because CXCR4-induced migration is required for DN thymocytes to migrate from the cortico-medullary junction where they first arrive in the thymus, to the sub-cortical zone where as DN3 cells they undergo β-selection (*Trampont et al., 2010*). Thus, the block in development of WNK1-deficient thymocytes may be due to defects in this stereotypical CXCR4-induced migration. To address this, we generated mixed bone marrow chimeras by reconstituting irradiated B6.SJL mice (CD45.1$^+$CD45.2$^-$) with an 80:20 mixture of marrow from B6.SJL mice and from either *Wnk1*$^{fl/+}$*Rag1*$^{-/-}$CD2-Cre or *Wnk1*$^{fl/fl}$*Rag1*$^{-/-}$CD2-Cre mice (CD45.1$^-$CD45.2$^+$). In the thymi of these animals, the RAG1-deficient hematopoietic cells can only develop as far as the DN3 stage, and we were able to analyze their distribution within normal thymic architecture generated by the excess of wild-type B6.SJL thymocytes. We found that WNK1-deficient DN3 cells accumulated in the sub-cortical zone with a similar frequency to WNK1-expressing cells and showed no excess accumulation at the cortico-medullary junction (*Figure 5F*). Thus, despite an in vitro migration defect, WNK1-deficient DN3 thymocytes migrate to the correct anatomical location in vivo, even when in competition with WT counterparts. Collectively these data indicate that the developmental block seen in WNK-1 deficient thymocytes is unlikely to be caused by defective localization within intra-thymic microenvironments.

To extend this, we made use of an in vitro thymocyte development assay in which DN thymocytes are cultured on OP9-DL1 cells (*Schmitt and Zúñiga-Pflücker, 2002*). In this system, over the course of 7d, thymocyte expansion and development occurs independently of the intra-thymic migration events seen in the thymus in vivo. Thus, we used it to evaluate if loss of WNK1 still affected thymic development even when migration was no longer relevant, by comparing the development of DN thymocytes from *Wnk1*$^{fl/+}$CD2-Cre and *Wnk1*$^{fl/fl}$CD2-Cre mice. As expected, we found that WNK1-expressing DN thymocytes expanded 100-fold in number and the majority differentiated into DP, 4SP and 8SP cells (*Figure 5G*). In contrast, WNK1-deficient cells did not increase in number and while most of them became DN4 cells (CD25$^-$CD44$^-$), almost none differentiated beyond the DN stage, resulting in a striking absence of DP thymocytes. Taken together, these results show that the block in T-cell development caused by WNK1 deficiency can be dissociated from its role in the regulation of thymocyte migration.

## WNK1 controls entry of thymocytes into S-phase

RNAseq analysis revealed a particularly strong cell-cycle signature at the 24 hr time point, though not at 0 or 3 hr (*Figure 3E*). While at 3 hr post-stimulation, induction of genes associated with progression from the early to late G1 phase of the cell cycle (CDK4, CDK6, CYCLIN D2 and CYCLIN D3) occurred normally (*Figure 6—figure supplement 1A*), at 24 hr genes encoding positive regulators of S-phase entry (CDK2, CYCLIN E1, CYCLIN E2, and E2F-, ORC- and MCM-family proteins) were downregulated in WNK1-deficient thymocytes, while negative regulators, such as *Cdkn1b*, *Cdkn1c* and *Fbwx7* were upregulated (*Figure 6A,B*, *Figure 6—figure supplement 1B*). To test whether S-phase entry is defective we measured DNA content of thymocytes from *Wnk1*$^{fl/+}$CD2-Cre and *Wnk1*$^{fl/fl}$CD2-Cre mice. We found that while 11% of both control and WNK1-deficient DN2 and DN3 thymocytes were in the S/G2/M phases of the cell cycle, at the DN4 stage 30% of control cells but

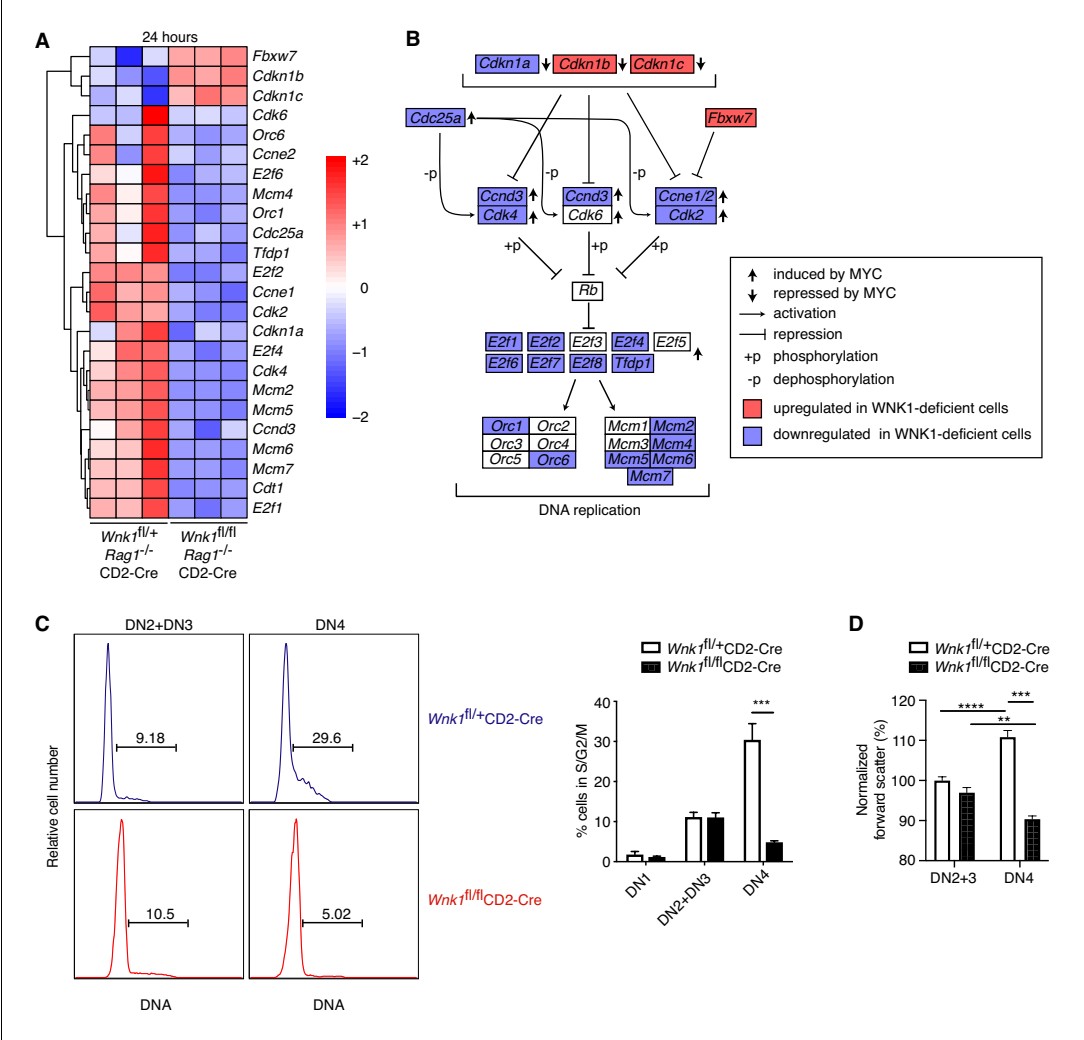

**Figure 6.** WNK1 controls entry into S-phase. (**A**) Analysis of RNAseq experiment described in *Figure 3A*, showing heatmap and hierarchical clustering of selected genes from the cell-cycle signature at 24 hr after injection of anti-CD3ε. (**B**) Canonical cell-cycle signaling diagram for the transition from G1 to S-phase. Genes colored in blue are downregulated at the 24 hr time point in samples from *Wnk1*<sup>fl/fl</sup>*Rag1*<sup>-/-</sup>CD2-Cre mice compared to *Wnk1*<sup>fl/+</sup>*Rag1*<sup>-/-</sup>CD2-Cre mice, genes in red are upregulated. Vertical arrows indicate genes induced or repressed by MYC (*Bretones et al., 2015*) (**C**) Left: flow cytometric analysis of DNA content in DN2+DN3 and DN4 thymocytes from *Wnk1*<sup>fl/+</sup>CD2-Cre and *Wnk1*<sup>fl/fl</sup>CD2-Cre mice. Gates show percentage of cells containing more than the diploid amount of DNA (S, G2 and M phases of cell cycle). Right: mean ± SEM percentage of cells in S/G2/M phases of the cell cycle based on gate shown on histograms. (**D**) Mean ± SEM forward scatter, a measure of cell size, measured by flow cytometry of DN2+DN3 and DN4 thymocytes from *Wnk1*<sup>fl/+</sup>CD2-Cre and *Wnk1*<sup>fl/fl</sup>CD2-Cre mice; data normalized to *Wnk1*<sup>fl/+</sup>CD2-Cre DN2+DN3 cells (set to 100%). **0.001 < p < 0.01, ***0.0001 < p < 0.001, ****p<0.0001. Significance was calculated by the Mann-Whitney test. Sample sizes: three (**A**) or five (**C**) of each genotype and nine *Wnk1*<sup>fl/+</sup>CD2-Cre and seven *Wnk1*<sup>fl/fl</sup>CD2-Cre mice (**D**). Data are from one experiment (**A**), or pooled from two (**C, D**). The online version of this article includes the following figure supplement(s) for figure 6:

**Figure supplement 1.** Loss of WNK1 affects expression of genes associated with S-phase, but not late G1.

only 5% of WNK1-deficient cells had progressed beyond the G1 phase (*Figure 6C*). Furthermore, loss of WNK1 abrogated the increase in cell size seen during the transition of DN3 cells to the DN4 stage, a hallmark of dividing cells (*Figure 6D*). Thus, WNK1 is required for DN4 thymocytes to enter the S/G2/M phases of the cell cycle.

## WNK1 is required for pre-TCR induced regulation of MYC

To gain further insight into this cell-cycle defect, we used a computational approach to identify transcription factors whose targets were most enriched among either all DEGs or only DEGs encoding

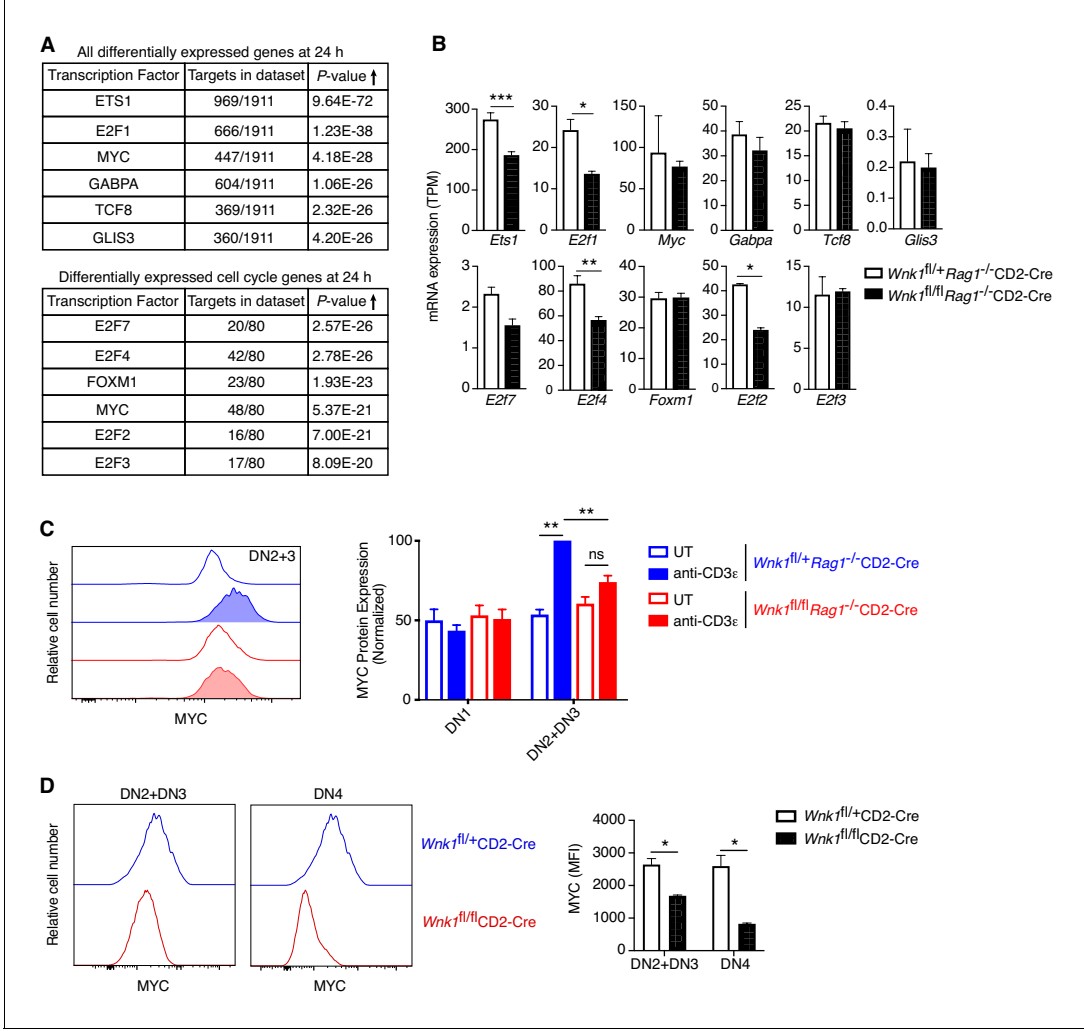

**Figure 7.** WNK1 is required for pre-TCR induced post-transcriptional regulation of MYC. (A, B) Analysis of RNAseq experiment described in *Figure 3A*. (A) Table shows top six transcription factors whose known target genes were most enriched among all significant DEGs at 24 hr or among only cell-cycle genes at 24 hr; results from Metacore transcription factor analysis. (B) Mean ± SEM expression of mRNAs for transcription factors identified in A. Note that graphs of *E2f1*, *E2f2*, *E2f4* and *E2f7* are duplicated from *Figure 6—figure supplement 1B*. (C) Left: flow cytometric analysis of MYC levels in DN2+DN3 thymocytes from *Wnk1*$^{fl/+}$*Rag1*$^{-/-}$CD2-Cre and *Wnk1*$^{fl/fl}$*Rag1*$^{-/-}$CD2-Cre mice either untreated (UT) or 48 hr after injection with anti-CD3ε. Right: mean ± SEM levels of MYC protein in DN1 and DN2+DN3 thymic subsets normalized to levels in DN2+DN3 cells from injected *Wnk1*$^{fl/+}$*Rag1*$^{-/-}$CD2-Cre control mice (set to 100%). (D) Left: flow cytometric analysis of MYC levels in DN2+DN3 or DN4 thymocytes from *Wnk1*$^{fl/+}$CD2-Cre and *Wnk1*$^{fl/fl}$CD2-Cre mice. Right: mean ± SEM levels of MYC in DN2+DN3 or DN4 thymocytes. MFI, mean fluorescence intensity. ns, not significant; *0.01 < p < 0.05, **0.001 < p < 0.01, ***p<0.001. Significance calculated by hypergeometric test (A), the Wald test (B), or the Mann-Whitney test (C, D). Sample sizes: three (A, B), five (C) or four (D) of each genotype. Data are from one experiment (A, B, D), or pooled from three (C).

The online version of this article includes the following figure supplement(s) for figure 7:

**Figure supplement 1.** Expression of anti-apoptotic genes in the absence of WNK1.

cell-cycle genes at 24 hr (*Figure 7A*) and extended the analysis to test the expression levels of the identified transcription factors themselves (*Figure 7B*). The transcription factor, whose known target genes were both most numerous and most highly enriched within the list of all DEGs at 24 hr was ETS1. Deletion of ETS1 impairs the development of DN3 into DP thymocytes because ETS1-deficient DN4 thymocytes are more likely to die (*Eyquem et al., 2004*). ETS1 regulates the expression of the anti-apoptotic *Bcl2A1* gene in endothelial cells (*Wei et al., 2009*), suggesting that it may perform a similar function in thymocytes. However, analysis of the five anti-apoptotic members of the BCL2-family that are expressed in thymocytes, showed that WNK1-deficient thymocytes express

substantially more *Bcl2* and *Bcl2l1* (*Figure 7—figure supplement 1A*). In addition, WNK1-deficient DN2+DN3 and DN4 thymocytes did not show increased levels of active Caspase-3 as would be expected if they were dying at a greater rate (*Figure 7—figure supplement 1B*). Thus, it is unlikely that loss of WNK1 perturbs cell survival at the β-selection checkpoint. Furthermore, loss of ETS1 does not affect proliferation of DN4 cells (*Eyquem et al., 2004*), thus reduced levels of ETS1 are not the main cause of the defect in WNK1-deficient thymocytes.

Another high scoring transcription factor was MYC, whose targets were also strongly enriched within all DEGs and the more limited cell-cycle gene set (*Figure 7A*). MYC regulates the expression of cell-cycle genes, including CDK-, CYCLIN-, and E2F-family proteins, many of which are dysregulated in WNK1-deficient thymocytes (*Figure 6B*; *Bretones et al., 2015*). Indeed, four direct MYC targets, E2F2, E2F3, E2F4, and E2F7, were identified among the transcription factors most likely to cause dysregulation of cell-cycle genes (*Figure 7A*). Furthermore, thymic development is blocked in the absence of MYC, because of a failure of DN thymocytes to proliferate (*Dose et al., 2006*; *Douglas et al., 2001*), similar to the phenotype in WNK1-deficient thymocytes. Thus, we hypothesized that WNK1 may be required for pre-TCR-induced proliferation in DN4 thymocytes because it regulates MYC activity.

*Myc* transcription in thymocytes is induced by Notch signaling (*Weng et al., 2006*), whereas pre-TCR signaling leads to increased MYC protein levels (*Dose et al., 2006*; *Mingueneau et al., 2013*). *Myc* mRNA levels were unaltered in WNK1-deficient DN3 thymocytes 24 hr after anti-CD3ε stimulation, implying that Notch regulation of *Myc* transcription is not affected in the absence of WNK1 (*Figure 7B*). The expression of another Notch target, *Hes1*, was also unaltered by the loss of WNK1 (*Figure 4B*), reinforcing the conclusion that WNK1 is not required for Notch function. In contrast, loss of WNK1 affected levels of MYC protein. Whereas anti-CD3ε stimulation resulted in increased MYC in control DN3 cells, no such rise was seen in WNK1-deficient cells (*Figure 7C*). Furthermore, analysis of *Wnk1*^fl/fl^CD2-Cre mice also showed reduced levels of MYC in both DN2+DN3 and DN4 thymocytes compared to control mice (*Figure 7D*). Thus, WNK1 is required for pre-TCR-induced post-transcriptional upregulation of MYC in DN thymocytes, and the failure to upregulate MYC in WNK1-deficient DN cells may account for their reduced proliferation and failure to differentiate into DP thymocytes.

## OXSR1 and STK39 are required for pre-TCR induced proliferation and regulation of MYC

We have previously shown that in CD4^+^ T cells, WNK1 transduces signals via the related OXSR1 and STK39 kinases and the SLC12A2 ion co-transporter (*Köchl et al., 2016*). To test if these proteins are also required for T-cell development, we evaluated thymocyte numbers in mice bearing a point mutation in *Stk39* (*Stk39*^T243A^) and *Oxsr1* (*Oxsr1*^T185A^). These residues are phosphorylated by WNK1 leading to the activation of STK39 and OXSR1, respectively and thus STK39-T243A and OXSR1-T185A can no longer be activated by WNK1. To analyze the effect of the *Stk39*^T243A^ mutation, we compared *Stk39*^T243A/T243A^ and control mice, whereas to assess the effect of the *Oxsr1*^T185A^ mutation we reconstituted irradiated RAG1-deficient mice with *Oxsr1*^T185A/T185A^ and control fetal liver cells, since this homozygous mutation is lethal in late gestation (*Rafiqi et al., 2010*). We found that thymocyte numbers in both mutant mouse strains were normal (*Figure 8A,B*), suggesting that individually WNK1 activation of either STK39 or OXSR1 is not required for thymocyte development.

However, since OXSR1 and STK39 are related kinases, they could be functionally redundant with each other. We were unable to test this possibility by simply combining the *Oxsr1*^T185A^ and *Stk39*^T243A^ mutations, since embryos homozygous for both mutations die early in gestation before fetal liver cells can be harvested to make radiation chimeras (not shown). Instead, we intercrossed mice with a floxed allele of *Oxsr1* (*Oxsr1*^fl^), the *Stk39*^T243A^ allele and RCE, the tamoxifen-inducible Cre to generate mice whose bone marrow cells we used to reconstitute the hematopoietic system of irradiated RAG1-deficient mice. Subsequent treatment of the mice with tamoxifen allowed us to compare development of control thymocytes expressing wild-type OXSR1 and STK39 (*Oxsr1*^+/+^*Stk39*^+/+^RCE), cells missing OXSR1 but expressing wild-type STK39 (*Oxsr1*^fl/fl^*Stk39*^+/+^RCE) or double mutant cells missing OXSR1 and expressing a mutant STK39 (*Oxsr1*^fl/fl^*Stk39*^T243A/T243A^RCE). In the double mutant animals there was a large reduction in DN4, ISP and DP thymocytes compared to control or single mutant mice (*Figure 8C*), demonstrating that OXSR1 and STK39 are required for development beyond the pre-TCR checkpoint, but that the kinases have redundant

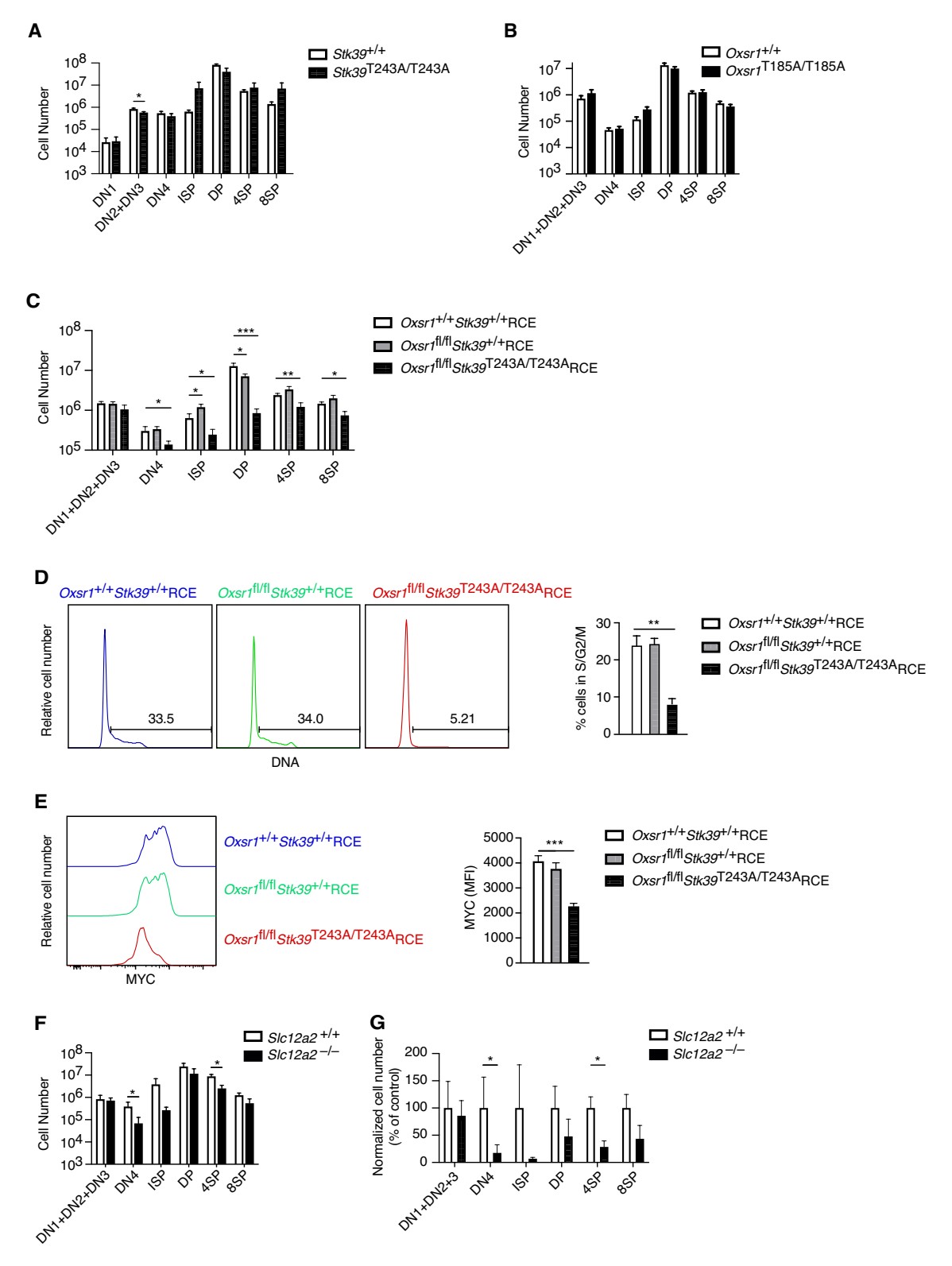

**Figure 8.** OXSR1 and STK39 are required for pre-TCR induced proliferation and regulation of MYC. (**A–C**) Mean ± SEM number of thymocytes at each developmental stage in $Stk39^{+/+}$ and $Stk39^{T243A/T243A}$ mice (**A**) or in $Rag1^{-/-}$ radiation chimeras reconstituted with fetal liver cells from $Oxsr1^{+/+}$ and $Oxsr1^{T185A/T185A}$ embryos (**B**) or reconstituted with bone marrow cells from $Oxsr1^{+/+}Stk39^{+/+}$RCE, $Oxsr1^{fl/fl}Stk39^{+/+}$RCE and $Oxsr1^{fl/fl}Stk39^{T243A/T243A}$RCE mice (**C**). The chimeric mice in C were analyzed 7 d after treatment with tamoxifen. Thymocyte subsets were identified by flow cytometry as shown in *Figure 8 continued on next page*

*Figure 8 continued*

*Figure 1A,E*. (D) Left: flow cytometric analysis of DNA content in DN4 thymocytes from *Rag1*$^{-/-}$ radiation chimeras reconstituted with bone marrow cells of the indicated genotypes. Gates show percentage of cells containing more than the diploid amount of DNA (S, G2, and M phases of cell cycle). Right: mean ± SEM percentage of cells in S/G2/M phases of the cell cycle based on gate shown on histograms. (E) Left: flow cytometric analysis of MYC levels in DN4 thymocytes from *Rag1*$^{-/-}$ radiation chimeras reconstituted with bone marrow cells of the indicated genotypes. Right: mean ± SEM levels of MYC in DN4 thymocytes. (F) Mean ± SEM number of thymocytes at each developmental stage in *Rag1*$^{-/-}$ radiation chimeras reconstituted with liver cells from *Slc12a2*$^{+/+}$ and *Slc12a2*$^{-/-}$ embryos. (G) Mean ± SEM number of cells in each thymocyte population in chimeras reconstituted with *Slc12a2*$^{-/-}$ fetal liver normalized to *Slc12a2*$^{+/+}$ control chimeras (set to 100%). *0.01 < p < 0.05, **0.001 < p < 0.01, and ***p<0.001. Significance calculated by Mann-Whitney test. Sample sizes: six (A, F, G) or eight (B) of each genotype, or eight *Oxsr1*$^{+/+}$*Stk39*$^{+/+}$RCE, and seven *Oxsr1*$^{fl/fl}$*Stk39*$^{+/+}$RCE and *Oxsr1*$^{fl/fl}$*Stk39*$^{T243A/T243A}$RCE mice (C–E). In all cases, data are pooled from two independent experiments.

function with each other. Furthermore, DN4 thymocytes deficient in both OXSR1 and STK39 had reduced entry into S-phase and reduced levels of MYC, similar to the phenotype seen in WNK1-deficient thymocytes (*Figure 8D,E*). Finally, we extended this to analysis of chimeras reconstituted with fetal livers from SLC12A2-deficient embryos and found their thymocytes to also be substantially reduced at the DN4, ISP, and 4SP stages, though the effect was not as large as that caused by loss of WNK1 (*Figure 8F,G*).

## Forced expression of MYC rescues the developmental block in WNK1-deficient thymocytes

The findings described above support a hypothesis in which reduced amounts of MYC in WNK1-deficient thymocytes were responsible for the defective pre-TCR checkpoint block. To directly test this, we performed MYC rescue experiments where irradiated RAG2-deficient mice were reconstituted with *Wnk1*$^{fl/fl}$RCE or control *Wnk1*$^{fl/+}$RCE bone marrow cells infected with retroviral vectors expressing MYC or MYC-T58A, a mutant that is degraded more slowly (*Gregory et al., 2003*; *Preston et al., 2015*; *Welcker et al., 2004a*; *Welcker et al., 2004b*). We used an empty vector as a control, and all retroviral vectors also expressed GFP to identify transduced cells (*Figure 9A*). The resultant chimeric mice were treated with tamoxifen to delete the *Wnk1* gene and analyzed 7 d later. As seen previously, loss of WNK1 resulted in a large reduction in the numbers of DN3b, DN4, ISP and DP thymocytes (*Figure 9B–D*). In both control and WNK1-deficient thymocytes infected with empty vector, the normalized % GFP$^+$ cells decreased from the DN3a to DN3b to DN4 subsets as thymocytes developed across the pre-TCR checkpoint (*Figure 9E,F*). In contrast, and compared to cells infected with empty vector, transduction of both control and WNK1-deficient thymocytes with MYC or MYC-T58A vectors resulted in a significant increase in the normalized % GFP$^+$ cells as they developed from DN3a to DN3b to DN4. Thus, cells overexpressing MYC demonstrate a selective advantage at the pre-TCR transition checkpoint. Importantly, this result implies that levels of endogenous MYC were limiting at this transition and are consistent with our hypothesis that the developmental block in WNK1-deficient thymocytes is caused by reduced levels of MYC.

Taken together, these results show that a WNK1-OXSR1-STK39-SLC12A2 pathway that regulates ion homeostasis plays a critical role at the pre-TCR checkpoint, by transducing signals that regulate thymocyte proliferation and differentiation into DP thymocytes, and that it may do so by controlling levels of MYC.

## Discussion

T-cell development in the thymus is a highly regulated process, and critical checkpoints at specific thymocyte stages remain poorly understood. We show here that WNK1 and its kinase activity are absolutely required for the transition of DN to DP thymocytes at the β-selection checkpoint. Specifically, in the absence of WNK1 there is a failure of the proliferative expansion of TCRβ$^+$ DN4 cells and subsequent differentiation into DP cells, hallmarks of β-selection. This developmental transition depends on signals from at least three receptors: pre-TCR, CXCR4, and NOTCH1 (*Carpenter and Bosselut, 2010*). NOTCH1 signaling appears to be unaffected by loss of WNK1, since WNK1-deficient DN thymocytes have normal expression of the NOTCH1-regulated *Hes1* and *Myc* genes. Instead, the developmental block in WNK1-deficient thymocytes is more likely caused by defective signaling from the pre-TCR, CXCR4 or both.

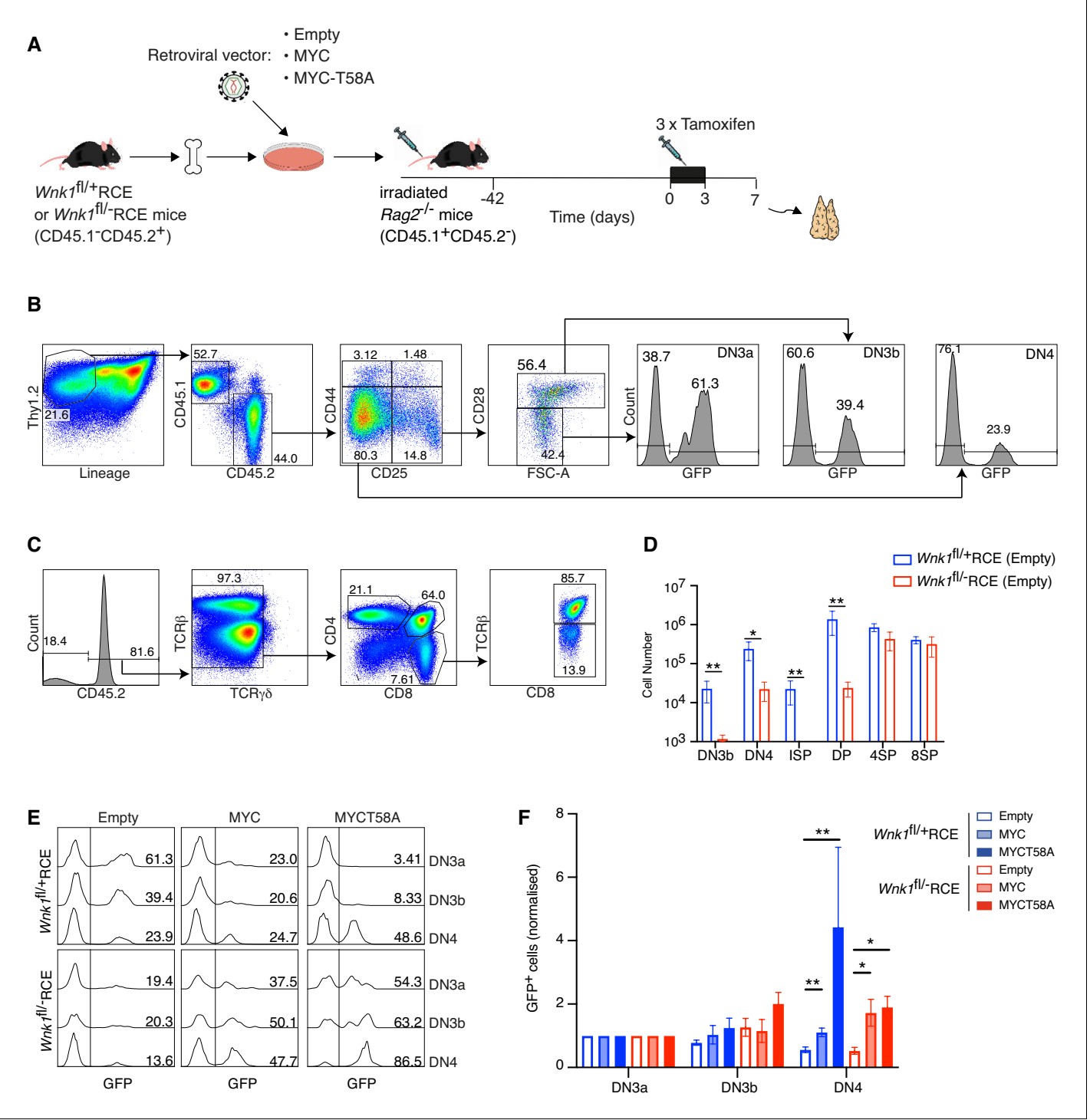

**Figure 9.** Ectopic expression of MYC promotes development of WNK1-deficient thymocytes through the pre-TCR checkpoint. (**A**) Bone marrow from *Wnk1*^fl/+RCE or *Wnk1*^fl/-RCE mice (CD45.1⁻CD45.2⁺) was infected in vitro with MIGR1 retroviral vectors expressing MYC, MYCT58A or no gene of interest (Empty); all vectors expressed GFP. Infected bone marrow was used to reconstitute irradiated RAG2-deficient mice (CD45.1⁺CD45.2⁻). 42 d later mice were treated with three daily injections tamoxifen and analyzed 7 d after start of tamoxifen treatment. (**B, C**) Flow cytometric analysis showing (**B**) gating of Thy1.2⁺Lineage⁻CD45.1⁻CD45.2⁺ thymocytes separated into DN3a (CD44⁻CD25⁺CD28⁻FSC^low), DN3b (CD44⁻CD25⁺CD28⁺FSC^high) and DN4 (CD44⁻CD25⁻CD28⁺) cells, within which GFP⁺ cells have been infected with the retroviral vector and (**C**) gating of CD45.2⁺TCRγδ⁻ thymocytes separated into ISP (CD4⁻CD8⁺TCRβ⁻), DP (CD4⁺CD8⁺), 4SP (CD4⁺CD8⁻) and 8SP (CD4⁻CD8⁺TCRβ⁺) cells. Numbers indicate percentage of cells in gates. (**D**) Mean ± SEM number of cells in thymocyte subsets of *Rag2*^-/- chimeras reconstituted with bone marrow from *Wnk1*^fl/+RCE or *Wnk1*^fl/-RCE mice retrovirally transduced with empty vector gated as in B and C. (**E**) Histograms showing GFP expression in DN3a, DN3b, and DN4 cells from chimeras
*Figure 9 continued on next page*

*Figure 9 continued*

reconstituted with *Wnk1*[fl/+]RCE or *Wnk1*[fl/-]RCE marrow that had been infected with indicated retroviral vectors. Numbers indicate percentage of GFP+ cells. (F) Graph of mean (± SEM) % GFP+ cells in DN3a, DN3b, and DN4 cells from chimeras reconstituted with *Wnk1*[fl/+]RCE or *Wnk1*[fl/-]RCE marrow that had been infected with indicated retroviral vectors, normalized to % GFP+ cells in DN3a cells of each chimera (set to 1). Significance calculated by Mann-Whitney test. *0.01 < p < 0.05, **p<0.01. Sample sizes: (D) five chimeras reconstituted with each genotype; (F) six chimeras reconstituted with *Wnk1*[fl/+]RCE marrow infected with empty vector or MIGR1-MYC, four chimeras reconstituted with *Wnk1*[fl/+]RCE marrow infected with MIGR1-MYCT58A, five chimeras reconstituted with *Wnk1*[fl/-]RCE marrow infected with empty vector or MIGR1-MYC and seven chimeras reconstituted with *Wnk1*[fl/-]RCE marrow infected with MIGR1-MYCT58A. Data are from one experiment.

Interestingly, the loss of WNK1 does not affect development of γδ thymocytes. The decision of whether DN thymocytes develop into the αβ or the γδ lineage is most likely driven by signal strength: a weak pre-TCR signal or a strong TCRγδ signal resulting in αβ or γδ development respectively (*Sumaria et al., 2019*). Our data imply that the stronger TCRγδ signal does not require WNK1.

Our results demonstrate that WNK1 transduces signals from the pre-TCR and CXCR4 in DN thymocytes. Furthermore, we showed that WNK1-deficient cells have increased anti-CD3ε- and CXCL12-induced LFA1-mediated adhesion and reduced CXCL12-induced migration. Thus, WNK1 is a negative regulator of adhesion and a positive regulator of migration in DN3 thymocytes. This mirrors our findings in CD4+ T cells, where we showed that WNK1 transduces signals from the TCR and CCR7 in CD4+ T cells to regulate both adhesion and migration (*Köchl et al., 2016*). WNK1 may be independently acting downstream of both the pre-TCR and CXCR4, but the two receptors may co-localize, with CXCR4 acting as a co-receptor for the pre-TCR (*Trampont et al., 2010*); other studies have shown that the TCR and CXCR4 physically associate to form a complex (*Kumar et al., 2006*). Thus, WNK1 may transduce signals from a pre-TCR–CXCR4 complex.

CXCR4-deficient mice show a strong developmental block at the β-selection checkpoint, similar to that seen in the absence of WNK1 (*Janas et al., 2010*; *Trampont et al., 2010*). This block was ascribed to mislocalization of DN3 thymocytes, with fewer cells reaching the sub-capsular zone (*Trampont et al., 2010*). Reduced CXCL12-induced migration of WNK1-deficient DN3 thymocytes in vitro suggested that mislocalization of DN3 cells may also account for the developmental arrest in WNK1-deficient thymi. However in vivo we found that WNK1-deficient DN3 thymocytes localized correctly to the sub-capsular zone and defective migration is unlikely to account for the developmental defect. This conclusion is further supported by the strong developmental arrest of WNK1-deficient thymocytes cultured on OP9-DL1 cells, a system where thymocyte development does not require migration. Since development of DP cells in this system requires CXCL12 produced by the OP9-DL1 cells, CXCR4 contributes to development through the β-selection checkpoint by more than just controlling thymocyte migration (*Janas et al., 2010*). Indeed, CXCR4, acting with the pre-TCR is required for proliferation of DN thymocytes as they undergo β-selection (*Trampont et al., 2010*).

Analysis of WNK1-deficient DN3 thymocytes responding to pre-TCR signals showed that the initial responses of the cells are largely normal. They increase rates of translation, transiently induce expression of *Nur77*, upregulate expression of *Cd27* and *Cd28*, downregulate expression of *Il2ra*, *Rag2* and *Ptcra*, and lose surface expression of CD25, phenotypically becoming DN4 thymocytes. Furthermore, after 3 hr of stimulation with anti-CD3ε, they progress into the late G1 phase of the cell cycle as indicated by upregulation of *Cdk6*, *Ccnd2,* and *Ccnd3*. However, they have a profound block in entry into S-phase 24 hr after stimulation, with reduced expression of *Cdk2*, *Ccne1*, and *Ccne2*, as well as genes encoding E2F-, ORC-, and MCM-family proteins. In agreement with this, many fewer WNK1-deficient DN4 thymocytes entered S/G2/M phases of the cell cycle. Since proliferation of DN thymocytes is required for their differentiation into DP cells (*Kreslavsky et al., 2012*), this proliferative defect also accounts for the failure of WNK1-deficient cells to develop into DP thymocytes.

MYC is upregulated at the β-selection checkpoint as thymocytes transit from the DN3 to DN4 stage and start to proliferate (*Dose et al., 2006*; *Mingueneau et al., 2013*). Importantly, we found that WNK1-deficient DN3 thymocytes failed to upregulate MYC in response to pre-TCR stimulation. Furthermore, many MYC target genes were dysregulated in the absence of WNK1, especially cell-cycle genes. MYC-deficiency results in a similar thymic phenotype to that seen in the absence of WNK1 – reduced proliferation of DN thymocytes, failure to increase in cell size and a substantial reduction of DP thymocytes (*Dose et al., 2006*; *Douglas et al., 2001*). Importantly, ectopic

expression of MYC provided a selective advantage for WNK1-deficient thymocytes developing across the pre-TCR checkpoint. Taken together, we propose that the reduced proliferation of WNK1-deficient DN4 thymocytes may be caused by failure to upregulate MYC, and hence reduced expression of genes required for S-phase entry.

The upregulation of MYC at the β-selection checkpoint is post-transcriptional – MYC protein increases despite constant *Myc* mRNA levels (*Mingueneau et al., 2013*). Loss of WNK1 affects this post-transcriptional regulation since mutant DN thymocytes had reduced MYC protein, but unchanged levels of *Myc* mRNA. MYC is regulated at the level of protein stability. It has a short half-life, estimated to be around 20 min, caused by rapid proteasomal degradation and its stability is regulated by phosphorylation; ERK phosphorylation stabilizes MYC, whereas GSK3β-dependent phosphorylation results in ubiquitination of MYC by E3-ligases such as FBXW7 and subsequent pro-teasomal degradation (*Dang, 2012*; *Gregory et al., 2003*; *Welcker et al., 2004a*; *Welcker et al., 2004b*; *Yada et al., 2004*). Notably, deletion of *Fbxw7* in the thymus results in high levels of MYC, increased proliferation and hypercellularity (*Onoyama et al., 2007*). Interestingly, the expression of *Fbxw7* is increased in WNK1-deficient DN3 thymocytes 24 hr after anti-CD3ε stimulation, which may contribute to destabilizing MYC.

Since loss of both OXSR1 and STK39 also leads to a block at the pre-TCR checkpoint, along with reduced MYC levels and decreased proliferation of DN4 thymocytes, we hypothesize that WNK1 transduces pre-TCR and CXCR4 signals via OXSR1 and STK39 to the upregulation of MYC, and hence to entry into S-phase and increased cell division. OXSR1 and STK39 phosphorylate multiple members of the SLC12A-family of ion co-transporters. Deficiency of SLC12A2, a $Na^+K^+Cl^-$ trans-porter, showed a block at the pre-TCR checkpoint, which was less complete than that seen in the absence of WNK1. This suggests that OXSR1 and STK39 may be transducing signals that regulate MYC via several SLC12A-family proteins, including SLC12A2. The WNK1-OXSR1-STK39-SLC12A2 pathway regulates ion uptake in kidney epithelial cells. Our results suggest that regulation of ion homeostasis may be essential for pre-TCR induced proliferation of thymocytes. Further studies will be needed to establish if it is the ion movement per se or the subsequent osmotic movement of water into the cell which is essential for thymic development.

Interestingly, among >1400 cancer cell lines, the expression of WNK1 is highest in T-cell acute lymphoblastic leukemia (T-ALL) lines, many of which originate from immature thymocytes (Cancer Cell Line Encyclopedia, CCLE https://portals.broadinstitute.org/ccle) (*Barretina et al., 2012*; *Tan et al., 2017*). Furthermore, T-ALL lines are the most sensitive to loss of WNK1, as measured by growth inhibition (CCLE) (*Tsherniak et al., 2017*), implying that WNK1 plays a key role in these thymically-derived tumors. Since T-ALL cells express high levels of MYC, which is essential for their proliferation (*Sanchez-Martin and Ferrando, 2017*), we hypothesize that WNK1 may contribute to the growth of T-ALL by maintaining high levels of MYC, making WNK1 a potential therapeutic target in this thymocyte-derived leukemia.

In summary we have shown that WNK1 is an essential regulator of early thymocyte development, where, during β-selection, it transduces signals from the pre-TCR and CXCR4 via OXSR1, STK39, and SLC12A2 that lead to upregulation of MYC, proliferation of DN4 thymocytes and differentiation into DP cells.

## Materials and methods

### Mice

Mice with a conditional allele of *Wnk1* containing loxP sites flanking exon 2 (*Wnk1*^tm1Clhu, *Wnk1*^fl), with a conditional allele of *Oxsr1* containing loxP sites flanking exons 9 and 10 (*Oxsr1*^tm1.1Ssy, *Oxsr1*^fl) with a transgenic Cre recombinase under the control of the human CD2 promoter (Tg(CD2-icre) 4Kio, CD2-Cre), with a kinase-inactive allele of *Wnk1* (*Wnk1*^tm1.1Tyb, *Wnk1*^D368A), with a tamoxifen-inducible Cre in the *ROSA26* locus (*Gt(ROSA)26Sor*^tm1(cre/ERT2)Thl, *ROSA26*^CreERT2, RCE), mice expressing OXSR1-T185A (*Oxsr1*^tm1.1Arte, *Oxsr1*^T185A), STK39-T243A (*Stk39*^tm1.1Arte, *Stk39*^T243A), and carrying loss of function alleles of *Slc12a2* (*Slc12a2*^tm1Ges *Slc12a2*^-, imported from MMRRC), *Rag2* (*Rag2*^tm1Fwa, *Rag2*^-) and *Rag1* (*Rag1*^tm1Mom, *Rag1*^-) have been described before (*de Boer et al., 2003*; *de Luca et al., 2005*; *Flagella et al., 1999*; *Köchl et al., 2016*; *Lin et al., 2011*; *Mombaerts et al., 1992*; *Rafiqi et al., 2010*; *Shinkai et al., 1992*; *Xie et al., 2009*). Mice with a

deleted allele of *Wnk1* (*Wnk1*[tm1.1Clhu], *Wnk1*[-]) were generated by crossing *Wnk1*[fl/+] mice with C57BL/6J.129S4-Tg(Prm-cre)70Og/Nimr mice that delete loxP-flanked alleles in the male germline (*O'Gorman et al., 1997*). All mice were bred and maintained on a C57BL/6J background, except RAG2-deficient mice which were maintained on a B6.SJL background. All breeding was initially carried out at the MRC National Institute for Medical Research and then at the Francis Crick Institute. All experiments were carried out under the authority of a Project Licence granted by the UK Home Office.

## Radiation chimeras

To generate radiation chimeras, either bone marrow cells were harvested from *Wnk1*[fl/+]RCE or *Wnk1*[fl/D368A]RCE mice or from *Oxsr1*[+/+]*Stk39*[+/+]RCE, *Oxsr1*[fl/fl]*Stk39*[+/+]RCE and *Oxsr1*[fl/fl]*Stk39*[T243A/T243A]RCE mice, or fetal livers were harvested from E14.5 embryos with the required genotypes (*Oxsr1*[+/+] and *Oxsr1*[T185A/T185A] or *Slc12a2*[+/-] and *Slc12a2*[-/-]). RAG1-deficient animals (5–8 weeks of age) were irradiated with 5Gy using a [137]Cs-source, and then reconstituted intravenously with at least $1 \times 10^6$ bone marrow cells/recipient or $0.5 \times 10^6$ fetal liver cells/recipient. All chimeric animals received Baytril in their drinking water (0.02%, Bayer Healthcare) for at least 4 weeks post-transplantation. If required, 8–20 weeks after reconstitution chimeric mice were injected intraperitoneally for 3 d with 2 mg/day of tamoxifen (Sigma) resuspended at 20 mg/ml in corn oil (Sigma) and analyzed 7 d after start of tamoxifen treatment.

## Ectopic expression of MYC in radiation chimeras

Mouse MYC cDNA was inserted into the *Eco*RI site of the MIGR1 retroviral vector (*Pear et al., 1998*) to generate MIGR1-MYC in which the MYC cDNA is followed by an internal ribosomal entry site and GFP (*Luo et al., 2005*). MIGR1-MYCT58A was generated by site directed mutagenesis from MIGR1-MYC. Infectious retroviruses were generated from these vectors by transfection into Plat-E cells (*Morita et al., 2000*) and used to infect bone marrow cells as previously described (*Schweighoffer et al., 2013*). Bone marrow cells from *Wnk1*[fl/+]RCE or *Wnk1*[fl/-]RCE mice on a C57BL/6J background (CD45.1[-]CD45.2[+]) were infected with MIGR1 (empty vector), MIGR1-MYC or MIGR1-MYCT58A retroviral vectors. RAG2-deficient mice on a B6.SJL background (CD45.1[+]CD45.2[-]) that had been irradiated with 5Gy using a [137]Cs-source were reconstituted intravenously with at least $1 \times 10^6$ infected bone marrow cells/recipient. Chimeric mice received Baytril in their drinking water for at least 4 weeks post-transplantation. 6 weeks after reconstitution chimeric mice were injected intraperitoneally for 3 d with 2 mg/day of tamoxifen and analyzed 7 d after start of tamoxifen treatment.

## Flow cytometry, antibodies, cytokines, and other materials

Flow cytometry was carried out using standard techniques with pre-titered antibodies. Antibodies for stimulations and flow cytometry against the following proteins were obtained from BioLegend, eBioscience, BD Biosciences, BD Pharmingen or Tonbo (clone names indicated in parentheses): B220 (RA3-6B2), CD3ε (2C11), CD4 (RM4-4 or GK1.5), CD8 (53–6.7), CD11b (M1/70), CD11c (N418 or HL3), CD19 (1D3 or MB19-1) CD25 (3C7 or PC-61.5 or 7D4), CD28 (E18), CD44 (IM7), CD45.1 (A20), CD45.2 (104), DX5, GR-1 (RB6-8C5), LFA-1 (M17/4), NK1.1 (PK136), PTCRA (2F5), TCRβ (H57-597), TCRγδ (UC7-13-D5), Thy1.2 (53–2.1). Further reagents: anti-active Caspase-3 (C92-605, BD Pharmingen), anti-CXCR4 (2B11, BD Pharmingen), anti-mouse IgG1-biotin (A85-1, BD Pharmingen), anti-MYC (Y69, Abcam), anti-pS325-OXSR1/pS383-STK39 (MRC-PPU), anti-pS235/pS236-S6 (D57.2.2E, Cell Signaling); 7AAD, Foxp3/Transcription factor staining buffer set (eBioscience), mouse CXCL12, mouse ICAM1-Fc (R and D systems), FxCycle Violet Stain, LIVE/DEAD NearIR (Thermo Fisher), Zombie Aqua (Biolegend). If not stated otherwise antibody dilutions were 1:200 for flow cytometry and thymocyte enrichment, and 1:1000 for immunoblotting experiments. If not otherwise stated antibodies to exclude lineage were a mixture of antibodies against the proteins B220, CD3ε, CD4, CD8, CD11b, CD11c, CD19, DX5, GR-1, NK1.1, TCRβ and TCRγδ. For the analysis of intracellular phospho-S6 levels single-cell suspensions of thymocytes were fixed in 2% PFA for 15 min at 4°C and then permeabilized with ice-cold 90% methanol over night at −20°C. Thymocytes were washed 2 x with PBS, containing 0.5% FCS before adding anti-pS235/pS236-S6 (1:50) and antibodies against surface markers in PBS with 0.5% FCS. Cells were analyzed by flow cytometry. For the analysis of

intracellular MYC or active Caspase-3, single-cell suspensions of thymocytes were first stained with antibodies against surface markers and then fixed for 20 min with BD Fix/Perm solution, before being washed 2x with BD Perm/Wash buffer and incubated with anti-MYC (1:400) or anti-active-Caspase-3-PE (1:50) in Perm/Wash buffer for 30 min. After two further washes with BD Perm/Wash buffer cells were analyzed by flow cytometry. For the analysis of DNA content single-cell suspensions of thymocytes were first stained with antibodies against surface markers and then fixed for 20 min with Fix/Perm buffer from the Foxp3 kit, before being washed 3x with Perm/Wash buffer from the kit and incubated in 7AAD or with FxCycle Violet Stain diluted 1:400 or 1:1000, respectively, into Perm/Wash buffer for 30 min. 7AAD and FxCycle Violet Stain fluorescence was analyzed by flow cytometry, with the fluorescence parameter set to linear; in all other cases, fluorescence was acquired on a logarithmic scale.

## Cell lines

OP9-DL1 and Platinum E (Plat-E) packaging cells were obtained from Cell Services of the Francis Crick Institute and were validated by Short Tandem Repeat profiling and tested for absence of mycoplasma contamination.

## RNA sequencing (RNAseq)

For RNAseq, $Wnk1^{fl/+}Rag1^{-/-}$CD2-Cre and $Wnk1^{fl/fl}Rag1^{-/-}$CD2-Cre mice were injected with anti-CD3ε (150 µg). Thymi were harvested 3, 24, and 48 hr later. Thymi from uninjected mice were used for the 0 hr time point. CD44$^-$ Thy1.2$^+$ thymocytes were sorted into Trizol (Life Technologies) using a Beckman Coulter MoFlo XDP cell sorter. RNA was purified using the RNEasy mini kit (Qiagen). Stranded polyA-enriched libraries were made using the KAPA mRNA HyperPrep Kit (Roche) according to the manufacturer's instructions and sequenced in the HiSeq 4000 (Illumina), collecting 28–73 million single-end reads of 75 bases per sample. Read trimming and adapter removal were performed using Trimmomatic (version 0.36) (*Bolger et al., 2014*). The RSEM package (version 1.2.31) (*Li and Dewey, 2011*), and STAR (version 2.5.2a) (*Dobin et al., 2013*) were used to align reads to the mouse genome (Ensembl GRCm38 release 86) and to obtain gene-level counts. For RSEM, all parameters were run as default except '–forward-prob' that was set to '0'. Differential expression analysis was carried out with DESeq2 package (*Love et al., 2014*). Genes were considered to be differentially expressed with adjusted p-value (padj) ≤0.05. For data processing in Metacore, only genes with an average expression of transcripts per million (TPM) >3 over all conditions were considered. All RNAseq data have been deposited in Gene Expression Omnibus (GEO) with accession number GSE136210.

For the re-analysis of RNAseq data from thymocyte subsets (*Hu et al., 2013*), Fastq files were obtained from the Gene Expression Omnibus (GSE48138). Gene-level TPM abundance estimates were calculated with RSEM using STAR to align reads against the NCBIM37 (mm9) genome assembly with Ensembl release 67 transcript annotations.

## Q-PCR for *Wnk1* mRNA

Up to $5 \times 10^4$ cells of each thymic population from thymi of control or WNK1-deficient animals were sorted directly into RLT lysis buffer, total RNA was extracted with an RNAEasy Plus Micro Kit (Qiagen) and cDNA was synthesized with a Superscript III kit (Life Technologies). Samples were analyzed on an ABI 7900 using a TaqMan gene expression assay (Life Technologies) spanning exon1 and exon 2 of *Wnk1*. Data was normalized to *Hprt1* and analyzed using the comparative threshold cycle method.

## Enrichment of DN thymocytes

DN thymocytes were enriched by negative selection. Single-cell suspensions were incubated with biotin-conjugated antibodies against lineage markers B220, CD3ε, CD4, CD8, CD11b, CD11c, CD19, DX5, GR-1, NK1.1, TCRβ, and TCRγδ, and were then washed and incubated with Streptavidin-conjugated magnetic beads (Dynabeads, Life Technologies) and cells bound to the beads removed according to the manufacturer's instructions.

## Stimulation of thymocyte development and signaling in RAG1-deficient mice

Development of double positive thymocytes was induced by a single intraperitoneal injection of 30 µg anti-CD3ε antibody into RAG1-deficient mice, followed by harvesting the thymus for flow cytometric analysis 4 d later. For all other experiments where pre-TCR signaling responses were being measured, RAG1-deficient mice were injected with 150 µg anti-CD3ε.

## MetaCore analysis

The lists of filtered differentially expressed genes were uploaded to MetaCore and analyzed with the Process Networks analysis and Transcription Factor analysis modules (MetaCore version 19.1 build 69600).

## Heatmaps

Selected genes from a number of enriched processes were used to generate a heatmap using the R package pheatmap. Genes were clustered using an Euclidean distance matrix and complete linkage clustering. Red indicates higher expression and blue indicates low expression relative to the mean expression of the gene across all samples.

## Protein synthesis assay

Mice were injected intraperitoneally with 150 µg anti-CD3ε or left untreated. 24 hr later thymi were harvested. Single-cell suspensions were prepared and either pre-treated with 10 µM cycloheximide for 30 min or not, before being incubated with 5 µM O-propargyl-puromycin (OPP) or not for 30 min at 37°C. Thymocytes were washed 2x in PBS 0.5% FCS, incubated with surface antibodies and fixed with 1% PFA for 15 min at room temperature. Cells were permeabilized with 0.5% Triton for 15 min at room temperature, washed 2x with PBS 0.5% FCS and then stained with the Click-it Plus OPP Protein Synthesis Kit Alexa Fluor 647 (C10458, Thermo Fisher) according to the manufacturer's instructions. Cells were washed 2x PBS 0.5% FCS and analyzed by flow cytometry.

## Adhesion assays

Binding of ICAM1 complexes to primary mouse thymocytes was analyzed as described (*Konstandin et al., 2006*). Soluble ICAM1-Fc-F(ab')$_2$ complexes were generated by diluting APC-labeled goat anti-human IgG F(ab')$_2$ fragments (109-135-098, Jackson Immunoresearch) 1:6.25 with ICAM1-Fc (200 µg/ml final) in HBSS and incubated for 30 min in HBSS at 4°C. Thymocytes were rested for 3 hr in IMDM, 5% FCS at 37°C, centrifuged and re-suspended in HBSS, 0.5% BSA. Each adhesion reaction (50 µl) contained $20 \times 10^6$ cells/ml, 25 µg/ml ICAM1 complex and the appropriate stimulus and was incubated at 37°C for the indicated times. Cells were fixed in PFA for 20 min and binding of ICAM1 complexes to $CD25^+CD44^-$ DN2+DN3 thymocytes analyzed by flow cytometry.

## Migration assays

Migration assays were carried out in 96-well Transwell plates, containing polycarbonate filters (5 µm pore size, Corning). Transwell filters were coated overnight with 500 ng/ml mouse ICAM1-Fc in PBS and blocked with PBS, 2% BSA for 2 hr. The receiver plate was filled with RPMI, 0.5% BSA, containing CXCL12 (100 ng/ml) or no chemokine, and $4 \times 10^4$ DN thymocytes in RPMI, 0.5% BSA were added to each well of the filter plate. After 5 hr the filter plate was removed, EDTA was added to each well (40 mM final concentration) and the cells were transferred to 96 well V-bottom plates, spun, re-suspended in PBS, 0.5% BSA, and analyzed on a flow cytometer. Percentage migration was calculated by dividing the number of cells that migrated through the filter by the total number of cells that had been added to each well.

## Immunoblotting

DN3 thymocytes from indicated mouse lines were enriched by negative depletion with anti-CD44 and rested for at least 90 min in IMDM, 5% FCS at 37°C. During the last 60 min, either DMSO or 10 µM WNK463 (HY-100626, Insight Biotechnology) was added to the cell suspensions. If required, cells were stimulated with CXCL12 (500 ng/ml) for 1 min. Subsequent immunoblotting analysis was performed as described previously (*Degasperi et al., 2014*; *Reynolds et al., 2002*). The following

antibodies were used for detection of proteins by Western blotting: anti-ERK2 (C-14, Santa Cruz), anti-pS325-OXSR1/pS383-STK39 (MRC-PPU).

## Histology

Lethally irradiated B6.SJL mice were reconstituted with an 80:20 mixture of bone marrow from B6. SJL (CD45.2$^-$) and either $Wnk1^{fl/+}Rag1^{-/-}$CD2-Cre or $Wnk1^{fl/fl}Rag1^{-/-}$CD2-Cre mice (both CD45.2$^+$). Thymic tissue was isolated from the chimeras 8–12 weeks after reconstitution, and frozen 7 µm sections were cut and the fixed in acetone. Sections were stained with the following antibodies: ERTR5 (a gift from Dr W van Ewijk, Riken Research Centre for Allergy and Immunology, Yokohama, Japan) and detected with goat anti-rat IgM coupled to AlexaFluor647, biotinylated anti-CD45.2 detected with streptavidin-AlexaFluor555, anti-CD25 coupled to FITC, anti-CD11b coupled to eFluor450, and anti-CD11c coupled to eFluor450. For confocal quantification, a minimum of three separate thymus sections from each mouse were analyzed at least 10 µm apart, and the frequency of CD45.2$^+$CD25$^+$-CD11b$^-$CD11c$^-$ cells was recorded in the subcapsular zone, cortex, and cortico-medullary junction (CMJ) in a set area as described (*Cowan et al., 2014*). The CMJ was defined by ERTR5 staining for medullary epithelial cells. Images were obtained using a Zeiss LSM 780 microscope and analyzed using Zeiss LSM software.

## In vitro thymocyte development

OP9-DL1 cells were maintained as previously described (*Ciofani and Zúñiga-Pflücker, 2005*). DN2 +DN3 thymocytes (Lineage$^-$, Thy1.2$^+$, CD25$^+$) were sorted and for each experimental condition, 25,000 cells were cultured with 10 ng/ml IL7 in 48-well plates that had been previously seeded with 10,000 irradiated OP9-DL1 cells. Cells were cultured for 7 d and then analyzed by flow cytometry.

## Statistical analysis

All statistical comparisons were carried out using the nonparametric two-tailed Mann-Whitney test, Hypergeometric test, Wald test, or two-way ANOVA. The specific analysis used is described in each Figure Legend, along with the number of times experiments were carried out and the number of samples (n numbers) for each experiment. N numbers always refer to biological repeats, not to technical repeats. Biological repeats were defined as analysis carried out on different animals, or on cells from different animals. Repeat measurements on cells from the same animal were defined as technical repeats. Only biological repeats were used for statistical analysis. No outliers were removed and no experiments were excluded.

## Acknowledgements

We thank George Kassiotis, Andreas Wack and Edina Schweighoffer for critical reading of the manuscript, Rob de Bruin, Daniel Pennington and Nital Sumaria for advice and Richard Mitter for help with bioinformatics analysis. We thank the Flow Cytometry, Advanced Sequencing, Cell Services and Biological Research Facility of the Francis Crick Institute for flow cytometry, RNA sequencing, provision of cell lines and animal husbandry respectively. We thank Chou-Long Huang, Dario Alessi and Sung-Sen Yang for mouse strains and Warren Pear and Michael Tomasson for retroviral vectors. VLJT was supported by the Biotechnology and Biological Sciences Research Council (grant BB/L00805X/1), by the UK Medical Research Council (Programme U117527252) and by the Francis Crick Institute which receives its core funding from Cancer Research UK (FC001194), the UK Medical Research Council (FC001194), and the Wellcome Trust (FC001194). GA was supported by the UK Medical Research Council (MR/N000919/1).

## Additional information

### Funding

| Funder | Grant reference number | Author |
| --- | --- | --- |
| Medical Research Council | U117527252 | Victor LJ Tybulewicz |
| Francis Crick Institute | FC001194 | Victor LJ Tybulewicz |

| Medical Research Council | FC001194 | Victor LJ Tybulewicz |
|---|---|---|
| Wellcome Trust | FC001194 | Victor LJ Tybulewicz |
| Cancer Research UK | FC001194 | Victor LJ Tybulewicz |
| Biotechnology and Biological Sciences Research Council | BB/L00805X/1 | Victor LJ Tybulewicz |
| Medical Research Council | MR/N000919/1 | Graham Anderson |

The funders had no role in study design, data collection and interpretation, or the decision to submit the work for publication.

## Author contributions

Robert Köchl, Conceptualization, Investigation, Writing - original draft, Writing - review and editing; Lesley Vanes, Harald Hartweger, Kathryn Fountain, Andrea White, Jennifer Cowan, Investigation; Miriam Llorian Sopena, Probir Chakravarty, Formal analysis; Graham Anderson, Supervision, Writing - review and editing; Victor LJ Tybulewicz, Conceptualization, Supervision, Funding acquisition, Writing - original draft, Writing - review and editing

## Author ORCIDs

Victor LJ Tybulewicz  https://orcid.org/0000-0003-2439-0798

## Ethics

Animal experimentation: All experiments were carried out under the authority of a Project Licence granted by the UK Home Office (PPL70/8843).

## Decision letter and Author response

Decision letter https://doi.org/10.7554/eLife.56934.sa1
Author response https://doi.org/10.7554/eLife.56934.sa2

# Additional files

## Supplementary files

• Supplementary file 1. RNAseq analysis of control and WNK1-deficient DN3 thymocytes following injection of anti-CD3ε. Analysis of RNAseq experiment described in *Figure 3A*, showing expression of all genes measured in TPM at all time points.

• Supplementary file 2. Differential gene expression in control and WNK1-deficient DN3 thymocytes following injection of anti-CD3ε. Analysis of RNAseq experiment described in *Figure 3A*, showing differential gene expression analysis with DESeq2, taking into account all genes.

• Supplementary file 3. Significant differential gene expression in control and WNK1-deficient DN3 thymocytes following injection of anti-CD3ε. Analysis of RNAseq experiment described in *Figure 3A*, showing differential gene expression analysis with DESeq2 as in *Supplementary file 2* but showing only genes that were statistically significantly differentially expressed (padj ≤0.05) and had an average expression value of TPM >3 over all conditions.

• Transparent reporting form

## Data availability

RNAseq data have been deposited in GEO under accession number GSE136210.

The following dataset was generated:

| Author(s) | Year | Dataset title | Dataset URL | Database and Identifier |
|---|---|---|---|---|
| Tybulewicz V, Köchl R, Llorian-Sopena M | 2019 | Analysis of anti-CD3e-induced transcriptional changes in WNK1-deficient thymocytes | https://www.ncbi.nlm.nih.gov/geo/query/acc.cgi?&acc=GSE136210 | NCBI Gene Expression Omnibus, GSE136210 |

The following previously published dataset was used:

| Author(s) | Year | Dataset title | Dataset URL | Database and Identifier |
|---|---|---|---|---|
| Gangqing H, Qing-song T, Suveena S, Fang Y, Thelma E, Stefan M, Jinfang Z, Keji Z | 2013 | Expression and regulation of lincRNAs during T cell development and differentiation | https://www.ncbi.nlm.nih.gov/geo/query/acc.cgi?acc=GSE48138 | NCBI Gene Expression Omnibus, GSE48138 |

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

# Appendix 1

**Appendix 1—key resources table**

| Reagent type (species) or resource | Designation | Source or reference | Identifiers | Additional information |
|---|---|---|---|---|
| Gene (*Mus musculus*) | *Wnk1* | Ensembl | ENSMUSG00000045962 | |
| Gene (*Mus musculus*) | *Oxsr1* | Ensembl | ENSMUSG00000036737 | |
| Gene (*Mus musculus*) | *Stk39* | Ensembl | ENSMUSG00000027030 | |
| Gene (*Mus musculus*) | *Slc12a2* | Ensembl | ENSMUSG00000024597 | |
| Gene (*Mus musculus*) | *Myc* | Ensembl | ENSMUSG00000022346 | |
| Genetic reagent (*Mus musculus*) | *Wnk1*$^{tm1Clhu}$; *Wnk1*$^{fl}$ | *Xie et al., 2009* doi:10.2353/ajpath.2009.090094 | RRID:MGI:4360972 | |
| Genetic reagent (*Mus musculus*) | *Wnk1*$^{tm1.1Clhu}$; *Wnk1*$^{-}$ | *Köchl et al., 2016* doi:10.1038/ni.3495 | RRID:MGI:6286269 | |
| Genetic reagent (*Mus musculus*) | *Oxsr1*$^{tm1.1Ssy}$; *Oxsr1*$^{fl}$ | *Lin et al., 2011* doi:10.1073/pnas.1107452108 | RRID:MGI:5297491 | |
| Genetic reagent (*Mus musculus*) | Tg(CD2-icre)4Kio; CD2Cre | *de Boer et al., 2003* doi:10.1002/immu.200310005 | RRID:MGI:2449947 | |
| Genetic reagent (*Mus musculus*) | *Wnk1*$^{tm1.1Tyb}$; *Wnk1*$^{D368A}$ | *Köchl et al., 2016* doi:10.1038/ni.3495 | RRID:MGI:6286266 | |
| Genetic reagent (*Mus musculus*) | *Gt(ROSA)26Sor*$^{tm1(cre/ERT2)Thl}$; *ROSA26*$^{CreERT2}$, RCE | *de Luca et al., 2005* doi:10.1172/JCI24059 | RRID:MGI:3701992 | |
| Genetic reagent (*Mus musculus*) | *Oxsr1*$^{tm1.1Arte}$; *Oxsr1*$^{T185A}$ | *Rafiqi et al., 2010* doi:10.1002/emmm.200900058 | RRID:MGI:5462099 | |
| Genetic reagent (*Mus musculus*) | *Stk39*$^{tm1.1Arte}$; *Stk39*$^{T243A}$ | *Rafiqi et al., 2010* doi:10.1002/emmm.200900058 | RRID:MGI:5462098 | |
| Genetic reagent (*Mus musculus*) | *Slc12a2*$^{tm1Ges}$; *Slc12a2*$^{-}$ | *Flagella et al., 1999* doi:10.1074/jbc.274.38.26946 | RRID:MGI:1935144 | |
| Genetic reagent (*Mus musculus*) | *Rag2*$^{tm1Fwa}$; *Rag2*$^{-}$ | *Shinkai et al., 1992* doi:10.1016/0092-8674(92)90029 C | RRID:MGI:1858556 | |

*Continued on next page*

*Appendix 1—key resources table continued*

| Reagent type (species) or resource | Designation | Source or reference | Identifiers | Additional information |
|---|---|---|---|---|
| Genetic reagent (*Mus musculus*) | *Rag1*tm1Mom; *Rag1*- | **Mombaerts et al., 1992** doi:10.1016/0092-8674(92)90030 G | RRID:MGI:1857241 | |
| Genetic reagent (*Mus musculus*) | C57BL/6J.129S4-Tg(Prm-cre)70Og | **O'Gorman et al., 1997** doi:10.1073/pnas.94.26.14602 | RRID:MGI:2388049 | |
| Cell line (*Mus musculus*) | OP9-DL1 | **Schmitt and Zúñiga-Pflücker, 2002** doi:10.1016/s1074-7613 (02)00474–0 | RRID:CVCL_B218 | |
| Cell line (*Homo-sapiens*) | Platinum E (Plat-E) | **Morita et al., 2000** doi:10.1038/sj.gt.3301206 | RRID:CVCL_B488 | Retroviral packaging cell line |
| Recombinant DNA reagent | pMIGR1 (plasmid) | **Pear et al., 1998** doi:10.1182/blood.V92.10.3780 | RRID:Addgene_27490 | Retroviral construct encoding GFP |
| Recombinant DNA reagent | pMIGR1-MYC (plasmid) | **Luo et al., 2005** doi:10.1182/blood-2005-02-0734 | | Retroviral construct encoding MYC and GFP |
| Recombinant DNA reagent | pMIGR1-MYCT58A (plasmid) | This paper | | Retroviral construct encoding MYC-T58A and GFP |
| Antibody | anti-B220-biotin (RA3-6B2) (Rat monoclonal) | Invitrogen | Cat# 13-0452-86; RRID:AB_466451 | FACS (1:200) |
| Antibody | anti-B220-FITC (RA3-6B2) (Rat monoclonal) | Biolegend | Cat# 103206; RRID:AB_312991 | FACS (1:200) |
| Antibody | anti-active Caspase 3-PE (C92-605) (Rabbit monoclonal) | BD Pharmingen | Cat# 550821; RRID:AB_393906 | FACS (1:50) |
| Antibody | anti-CD3ε-biotin (2C11) (Armenian hamster monoclonal) | Biolegend | Cat# 100304; RRID:AB_312669 | FACS (1:200) |
| Antibody | anti-CD3ε-FITC (2C11) (Armenian hamster monoclonal) | eBioscience | Cat# 11-0031-85; RRID:AB_464883 | FACS (1:200) |
| Antibody | anti-CD3ε purified, endotoxin low (2C11) (Armenian hamster monoclonal) | Tonbo | Cat# 40–0031 M001; RRID:AB_464883 | IV injection |
| Antibody | anti-CD4-biotin (RM4-4) (Rat monoclonal) | eBioscience | Cat# 13-0043-85; RRID:AB_466334 | FACS (1:200) |
| Antibody | anti-CD4-FITC (GK1.5) (Rat monoclonal) | eBioscience | Cat# 11-0041-86; RRID:AB_464894 | FACS (1:200) |
| Antibody | anti-CD4-BV421 (RM4-5) (Rat monoclonal) | Biolegend | Cat# 100544; RRID:AB_11219790 | FACS (1:200) |
| Antibody | anti-CD4-PerCP (RM4-5) (Rat monoclonal) | Biolegend | Cat# 100538; RRID:AB_893325 | FACS (1:200) |
| Antibody | anti-CD4-APC-CY7 (GK1.5) (Rat monoclonal) | BD Pharmingen | Cat# 552051; RRID:AB_394331 | FACS (1:200) |
| Antibody | anti-CD8-biotin (53–6.7) (Rat monoclonal) | Biolegend | Cat# 100704; RRID:AB_312743 | FACS (1:200) |

*Continued on next page*

*Appendix 1—key resources table continued*

| Reagent type (species) or resource | Designation | Source or reference | Identifiers | Additional information |
|---|---|---|---|---|
| Antibody | anti-CD8-FITC (53–6.7) (Rat monoclonal) | Biolegend | Cat# 100706; RRID:AB_312745 | FACS (1:200) |
| Antibody | anti-CD8-BV421 (53–6.7) (Rat monoclonal) | Biolegend | Cat# 100738; RRID:AB_11204079 | FACS (1:200) |
| Antibody | anti-CD8-APC (53–6.7) (Rat monoclonal) | eBioscience | Cat# 17-0081-83; RRID: AB_469336 | FACS (1:200) |
| Antibody | anti-CD8-PE-CY7 (53–6.7) (Rat monoclonal) | Biolegend | Cat# 100722; RRID:AB_312761 | FACS (1:200) |
| Antibody | anti-CD11b-biotin (M1/70) (Rat monoclonal) | Biolegend | Cat# 101204; RRID:AB_312787 | FACS (1:200) |
| Antibody | anti-CD11b-FITC (M1/70) (Rat monoclonal) | Biolegend | Cat# 101206; RRID:AB_312789 | FACS (1:200) |
| Antibody | anti-CD11c-biotin (N418) (Armenian Hamster monoclonal) | Biolegend | Cat# 117304; RRID:AB_313773 | FACS (1:200) |
| Antibody | anti-CD11c-FITC (HL3) (Armenian Hamster monoclonal) | BD Pharmingen | Cat# 553801; RRID:AB_395060 | FACS (1:200) |
| Antibody | anti-CD19-biotin (1D3) (Rat monoclonal) | BD Pharmingen | Cat# 553784; RRID:AB_395048 | FACS (1:200) |
| Antibody | anti-CD19-FITC (MB19-1) (Mouse monoclonal) | eBioscience | Cat# 11-0191-85; RRID: AB_464966 | FACS (1:200) |
| Antibody | anti-CD25-eF450 (PC-61.5) (Rat monoclonal) | eBioscience | Cat# 48-0251-82; RRID: AB_10671550 | FACS (1:200) |
| Antibody | anti-CD25-FITC (7D4) (Rat monoclonal) | BD Pharmingen | Cat# 553072; RRID:AB_394604 | FACS (1:200) |
| Antibody | anti-CD25-PE (3C7) (Rat monoclonal) | BD Pharmingen | Cat# 553075; RRID:AB_394605 | FACS (1:200) |
| Antibody | anti-CD28-PE-CY7 (E18) (Mouse monoclonal) | Biolegend | Cat# 122014; RRID:AB_604079 | FACS (1:200) |
| Antibody | anti-CD44-PE (IM7) (Rat monoclonal) | eBiosciene | Cat# 12-0441-83; RRID: AB_465665 | FACS (1:200) |
| Antibody | anti-CD44-APC (IM7) (Rat monoclonal) | eBioscience | Cat# 17-0441-83; RRID: AB_469391 | FACS (1:200) |
| Antibody | anti-CD44-APC-eF780 (IM7) (Rat monoclonal) | eBioscience | Cat# 47-0441-82; RRID: AB_1272244 | FACS (1:200) |
| Antibody | CD45.1-PerCP-Cy5.5 (A20) (Mouse monoclonal) | eBioscience | Cat# 45-0453-82; RRID: AB_1107003 | FACS (1:200) |
| Antibody | CD45.2-APC-eF780 (104) (Mouse monoclonal) | eBioscience | Cat# 47-0454-82; RRID: AB_1272175 | FACS (1:200) |
| Antibody | anti-CXCR4-PE (2B11) | BD Pharmingen | Cat# 12-9991-82; RRID: AB_891391 | FACS (1:100) |
| Antibody | anti-DX5-biotin (Rat monoclonal) | eBioscience | Cat# 13-5971-85; RRID: AB_466826 | FACS (1:200) |
| Antibody | anti-DX5-FITC (Rat monoclonal) | Biolegend | Cat# 108906; RRID:AB_313413 | FACS (1:200) |
| Antibody | anti-GR-1-biotin (RB6-8C5) (Rat monoclonal) | Biolegend | Cat# 108404; RRID:AB_313369 | FACS (1:200) |
| Antibody | anti-LFA-1-PE (M17/4) (Rat monoclonal) | BD Bioscience | Cat# 12-0111-82; RRID: AB_465544 | FACS (1:200) |

*Continued on next page*

*Appendix 1—key resources table continued*

| Reagent type (species) or resource | Designation | Source or reference | Identifiers | Additional information |
|---|---|---|---|---|
| Antibody | anti-mouse IgG1-biotin (A85-1) (Rat monoclonal) | BD Pharmingen | 553441; RRID:AB_394861 | FACS (1:100) |
| Antibody | anti-MYC-AF647 (Y69) (Rabbit monoclonal) | Abcam | Cat# ab190560; RRID:AB_2876372 | FACS (1:400) |
| Antibody | anti-NK1.1-biotin (PK136) (Mouse monoclonal) | eBioscience | Cat# 13-5941-85; RRID:AB_466805 | FACS (1:200) |
| Antibody | anti-NK1.1-FITC (PK136) (Mouse monoclonal) | Biolegend | Cat# 108706; RRID:AB_313393 | FACS (1:200) |
| Antibody | anti-pS325-OXSR1/pS383-STK39 (sheep polyclonal) | MRC-PPU | Cat# S670B; RRID:AB_2876373 | IB (1 µg/ml) |
| Antibody | anti-PTCRA (2F5) (Mouse monoclonal) | BD Pharmingen | Cat# 552407; RRID:AB_394381 | FACS (1:100) |
| Antibody | anti-pS235/pS236-S6-AF488 (D57.2.2E) (Rabbit monoclonal) | Cell Signaling | Cat# 5317; RRID:AB_10694920 | FACS (1:50) |
| Antibody | anti-TCRβ-bio (H57-597) (Armenian Hamster monoclonal) | eBioscience | Cat# 13-5961-85; RRID:AB_466820 | FACS (1:200) |
| Antibody | anti-TCRβ-FITC (H57-597) (Armenian Hamster monoclonal) | Biolegend | Cat# 109206; RRID:AB_313429 | FACS (1:200) |
| Antibody | anti-TCRβ-APC (H57-597) (Armenian Hamster monoclonal) | Tonbo | Cat# 20–5961 U100, RRID:AB_2621612 | FACS (1:200) |
| Antibody | anti-TCRγδ-biotin (UC7-13D5) (Armenian Hamster monoclonal) | eBioscience | Cat# 13-5811-82; RRID:AB_466684 | FACS (1:200) |
| Antibody | anti-TCRγδ-FITC (UC7-13D5) (Armenian Hamster monoclonal) | Biolegend | Cat# 107504; RRID:AB_313313 | FACS (1:200) |
| Antibody | anti-TCRγδ-PE (GL3) (Armenian Hamster monoclonal) | eBioscience | Cat# 12-5711-82; RRID:AB_465934 | FACS (1:200) |
| Antibody | anti-Thy1.2-BV605 (53–2.1) (Rat monoclonal) | Biolegend | Cat# 140317; RRID:AB_11203724 | FACS (1:500) |
| Antibody | anti-Thy1.2-FITC (53–2.1) (Rat monoclonal) | Biolegend | Cat# 140304; RRID:AB_10642812 | FACS (1:500) |
| Antibody | anti-Thy1.2-APC (53–2.1) (Rat monoclonal) | eBioscience | Cat# 17-0902-83; RRID:AB_469423 | FACS (1:500) |
| Peptide, recombinant protein | Streptavidin-PE | Biolegend | Cat #: 405204 | FACS (1:200) |
| Peptide, recombinant protein | Streptavidin-PerCP | Biolegend | Cat #: 405213 | FACS (1:200) |
| Peptide, recombinant protein | mouse CXCL12 | R and D systems | Cat # 460-SD-010/CF | (500 ng/ml) |
| Peptide, recombinant protein | mouse ICAM1-Fc | R and D systems | Cat # 796-IC-050 | Transwell assay (500 ng/ml); Adhesion Assay (25 µg/ml) |

*Continued on next page*

*Appendix 1—key resources table continued*

| Reagent type (species) or resource | Designation | Source or reference | Identifiers | Additional information |
|---|---|---|---|---|
| Commercial assay or kit | Foxp3/Transcription factor staining buffer set | eBioScience | Cat # 00-5523-00 | |
| Chemical compound, drug | Tamoxifen | Millipore-Sigma | Cat # T5648 | |
| Software, algorithm | Graphpad Prism 8 | Graphpad | RRID:SCR_002798 | |
| Software, algorithm | Flowjo 10 | Flowjo | RRID:SCR_008520 | |
| Other | 7AAD | Merck | Cat # SML1633 | |
| Other | FxCycle Violet Stain | Thermo Fisher | Cat # F10347 | (1:1000) |
| Other | LIVE/DEAD NearIR | Thermo Fisher | Cat # L10119 | (1:500) |
| Other | Zombie Aqua | Biolegend | Cat # 423102 | (1:500) |

