## [Decision Letter]

**Acceptance summary:**

The manuscript addresses the molecular pathways used by the pre-TCR, a key molecular checkpoint controlling the development of early T cells in the thymus. The WNK1-OXSR1-STK39-SLC12A2 signaling axis was specifically found to be used by the pre-TCR for the post-transcriptional upregulation of MYC and the subsequent proliferation and differentiation of developing T cells.

**Decision letter after peer review:**

Thank you for submitting your article "Critical role of WNK1 in MYC-dependent early thymocyte development" for consideration by *eLife*. Your article has been reviewed by three peer reviewers, and the evaluation has been overseen by a Reviewing Editor and Satyajit Rath as the Senior Editor. The following individuals involved in review of your submission have agreed to reveal their identity: Juan Carlos Zuniga-Pflucker (Reviewer #2); David Wiest (Reviewer #3).

The reviewers have discussed the reviews with one another and the Reviewing Editor has drafted this decision to help you prepare a revised submission.

As the editors have judged that your manuscript is of interest, but as described below that additional experiments are required before it is published, we would like to draw your attention to changes in our revision policy that we have made in response to COVID-19 (https://elifesciences.org/articles/57162). First, because many researchers have temporarily lost access to the labs, we will give authors as much time as they need to submit revised manuscripts. We are also offering, if you choose, to post the manuscript to bioRxiv (if it is not already there) along with this decision letter and a formal designation that the manuscript is 'in revision at *eLife*'. Please let us know if you would like to pursue this option. (If your work is more suitable for medRxiv, you will need to post the preprint yourself, as the mechanisms for us to do so are still in development.)

Based on the discussions between reviewers and the individual reviews below, we would like to inform you that a revision would be most welcome if it provides, at the very least:

1) The 'MYC rescue' experiment via ectopic expression of MYC in OP9-DL cultures.

2) Description of the effects of WNK1 mutation on γδ T cell development, given that they do not appear to be as dependent upon proliferation.

3) Essential information such as the numbers of mice that were analyzed.

Although the other requests for additional data in the reviews are sound, they might be too difficult to perform under the present circumstances. Nonetheless, the full reviews are being provided below for your reference and use in revision.

Reviewer #1:

In this manuscript, Kochl et al. demonstrate that WNK1 is required during early T cell development using genetic mouse models and claim a regulatory axis that controls pre-TCR selection. Unfortunately, the paper has issues regarding the data presentation and interpretation as a result of unclear definitions of the precise developmental stages. This leads to conflicting data and renders it difficult to understand the precise value of the work. In addition, the direct evidence for the proposed axis is insufficiently demonstrated. Therefore, I believe that a major reanalysis using more appropriate markers and with inclusion of rescue experiments would be required to support the claims of the authors.

1) There is a lack in clarity and accuracy for the dissection and identification of the stages of early thymocyte development (in the Introduction and experimental data in the manuscript) which is important for the interpretation of the data. This is already the case for the DN1 and DN2 stages which are poorly defined in light of the current state-of-the-art (DN1/ETP and DN2a/DN2b, Yui et al., 2010), but this is particularly an issue for understanding the impact at β-selection which now is confusingly associated with DN2/DN3 to DN4 as well as DN4 to ISP and DP stages. This renders the interpretation of the data extremely difficult and inconsistent despite the fact that β-selection has been clearly defined by various labs (Hayday using size, Rothenberg using CD27 and Hodes using CD28). Figure 3B suggests that there is no impact of β-selection in fact since the DN3 to DN4 transition there seems unaffected. However, the DN4 subsets also need a clear and better definition since this is a mixture of cells if insufficient markers are included in the panel. This fraction may include γδ T cells and other lineages. This is also important for clearly understanding the frequency of TCRβ expressing thymocytes in DN3 and DN4 stages (Figure 2A). Thus, it is extremely difficult to make clear conclusions on the precise developmental stage that is impacted by WNK1. This needs extensive revision in my opinion.

2) Another major issue relates to the choice of the mouse model for conditional deletion of *Wnk1*. The authors use hCD2-iCre to mediate deletion of floxed *Wnk1* gene and state that this deletion started at the DN2 stage. This is not in agreement with literature on this mouse model since hCD2-iCre expression has been detected in immature (ETP for T; pro-B for B lineage) and mature B/T cells, and also in bone marrow lymphoid progenitors (CLPs), natural killer cells and dendritic cells (Siegemund et al., 2015). In fact, the authors show in Figure 1D that significant reduction of *Wnk1* mRNA expression was detected at DN1 stage of *Wnk1*^fl/fl^/hCD2-iCre thymocytes compared to the control. So, the authors need to consider the pleiotropic effects of conditional deletion of *Wnk1* using hCD2-iCre mice which might be particularly important in relation to intrathymic DC development. Cpa3-Cre mice would allow T-specific deletion of *Wnk1* (Feyerabend, Immunity. 2009). Alternatively, the authors could perform knockdown of *Wnk1* in sorted wild type thymocytes and performed in-vitro co-culture assay to demonstrate the similar developmental defects observed in the mouse model.

3) There appear to be some inconsistencies in the data. In Figure 1A, the DN fraction of *Wnk1*-deleted thymi is much higher compared to the control (1.77% in *Wnk1*^fl/+^/hCD2-iCre vs. 55.8% in *Wnk1*^fl/fl^/hCD2-iCre). However, this is inconsistent with Figure 1B/C where none of the DN subsets have increased cell numbers or frequencies. In general, to understand the precise impact on cellularity and for clear interpretation of Figure 1, it is important to provide the data for total thymic cellularity. In Figure 2, the authors shown that DN4 cells are reduced in the KO. However, in Figure 3B, the authors show that there is no blockade or arrest in development of DN3 cells to DN4 stage. This is also inconsistent with the data in Figure 1B-C and may relate to the unclear definitions of the developmental stages.

4) It is not clear to me how many mice were analyzed and how the data is presented. For instance, the legend of Figure 1 states that data are from one of two experiments but according to the reporting form this should be more given that 5-7 mice were analyzed according to that same legend.

5) In general, there is insufficient direct evidence for the proposed regulatory axis as proposed by the authors. The data mainly shows indirect relationships and the paper would strongly benefit from functional rescue experiments to support these strong conclusions.

Reviewer #2:

The work by Köchl investigates the role of WNK1, a S/T kinase, during T cell development, by performing a set of elegant and beautifully executed experiments, which support the conclusion that WNK1 is necessary for the differentiation of thymocytes from the early CD4-CD8- (DN) to the CD4^+^CD8^+^ stage of T cell differentiation. The authors establish a signaling paradigm in which WNK1 appears to function downstream of preTCR/CXCR4 signaling and acts on OXSR1 and STK39 to somehow affect the post-translational stability of MYC, which is known to play a critical role at the DN to DP transition. Using elegant genetic approaches and adoptive transfer experiments, the intrinsic requirement for WNK1 function is clearly established, and the use of in vitro cultures separates the effect on cell adhesion and chemokine signaling from the differentiation block. Using an in vivo model of pre-T cell receptor signaling, RNAseq analyses reveals that proliferation and protein synthesis regulation are affected in the absence of WNK1. Overall, the findings are exciting and provide important insights as to the role of WNK1 signaling in T cell development are revealed by the work, which are major contributions to our current understanding.

1) The authors clearly establish that MYC protein levels in thymocytes are lower in the absence of WNK1, however it would be important to show whether ectopic expression of MYC could lead the rescue of the proliferation and/or differentiation defect. This can be easily performed, as per Wong et al. (PMID: 22649105), a related paper that examined the role of pre-TCR, Notch and MYC during T cell development.

2) The clear effect on pre-TCR signaling and the block in the DN to DP differentiation begs the question as to whether γ/δ-T cell differentiation is similarly affected or not, which would position WNK1 as uniquely required for α/β-T lineage differentiation or as shared signaling axis for both lineages. Simply showing whether γδ T cells are present in the thymus and peripheral tissues would directly address this point.

3) It curious that the authors did not directly address whether the absence of WNK1 in DN cells has an effect on any aspects of ion transport.

Reviewer #3:

The study by Kochl et al. explores the role of the Ser/Thr kinase WNK1 in regulating early T cell development. *Wnk1* function has been implicated in numerous cellular processes, including controlling the balance of migration vs. adhesion of mature T cells; however, its role in T cell development had not previously been assessed. The authors provide compelling evidence that *Wnk1* function is required for early thymocytes to traverse the β-selection checkpoint, proliferate and differentiate to the CD4^+^CD8^+^ stage. While defects in adhesion and migration of *Wnk1*-deficient thymocytes are noted, the impaired development appears to result from other *Wnk1*-regulated events and it is associated with impaired MYC induction. The authors propose that the failure to induce MYC results from attenuation of the activity of OXSR1 and STK39 kinases, which are necessary to induce the sodium/potassium/chloride ion transporter, SLC12A2, upon which MYC induction depends. While the link between *Wnk1* and the OXSR1/STK39/SLC12A2 axis had been described in other cells, it's role in MYC regulation and T cell development was not previously appreciated. Consequently, the authors provide support for a new role of this axis in controlling T cell development at least in part through effects on an ion channel.

This study is well designed, executed, and thoughtfully interpreted and brings new understanding to signaling axes controlling T cell development. There are however, a few issues that would strengthen the manuscript if addressed.

1) The important role played by STK39 is somewhat remarkable given how little appears to be expressed in early thymocytes. Could the authors please comment on this?

2) It would be useful for the authors to measure by flow the levels of MYC in the SLC12A2-ko thymocytes.

3) While the role of MYC in controlling cell size, cell proliferation, and thymocyte development is well documented, it would be a nice finishing touch to test whether ectopic expression of MYC would rescue the developmental defects seen in the *Wnk1*-deficient thymocytes, if only in vitro in OP9 cultures.

---

## [Author Response]

Based on the discussions between reviewers and the individual reviews below, we would like to inform you that a revision would be most welcome if it provides, at the very least:1) The ‘MYC rescue' experiment via ectopic expression of MYC in OP9-DL cultures.

We have carried out an in vivo rescue experiment in mice, which showed that ectopic expression of MYC or MYCT58A confers a selective advantage to both control and WNK1-deficient thymocytes developing across the pre-TCR checkpoint, consistent with our hypothesis that amounts of MYC are limiting in the absence of WNK1. This data is presented in a new Figure 9.

2) Description of the effects of WNK1 mutation on γδ T cell development, given that they do not appear to be as dependent upon proliferation.

We have added new data on the numbers of γδ thymocytes to Figure 1B, C. The loss of WNK1 does not affect their numbers.

3) Essential information such as the numbers of mice that were analyzed.

We have ensured that the numbers of mice analysed and the numbers of experiments are indicated on each figure legend, changing wording where this had been unclear.

Although the other requests for additional data in the reviews are sound, they might be too difficult to perform under the present circumstances. Nonetheless, the full reviews are being provided below for your reference and use in revision.

As described below, and despite limited access to the lab because of the COVID19 pandemic, we have been able to address a number of the additional points raised by the reviewers through the addition of extra data and by amending text.

Reviewer #1:In this manuscript, Kochl et al. demonstrate that WNK1 is required during early T cell development using genetic mouse models and claim a regulatory axis that controls pre-TCR selection. Unfortunately, the paper has issues regarding the data presentation and interpretation as a result of unclear definitions of the precise developmental stages. This leads to conflicting data and renders it difficult to understand the precise value of the work. In addition, the direct evidence for the proposed axis is insufficiently demonstrated. Therefore, I believe that a major reanalysis using more appropriate markers and with inclusion of rescue experiments would be required to support the claims of the authors.1) There is a lack in clarity and accuracy for the dissection and identification of the stages of early thymocyte development (in the Introduction and experimental data in the manuscript) which is important for the interpretation of the data. This is already the case for the DN1 and DN2 stages which are poorly defined in light of the current state-of-the-art (DN1/ETP and DN2a/DN2b, Yui et al., 2010), but this is particularly an issue for understanding the impact at β-selection which now is confusingly associated with DN2/DN3 to DN4 as well as DN4 to ISP and DP stages. This renders the interpretation of the data extremely difficult and inconsistent despite the fact that β-selection has been clearly defined by various labs (Hayday using size, Rothenberg using CD27 and Hodes using CD28).

We have edited the Introduction to clearly define DN1/ETP and DN2 cells as c-Kit+ (CD117+) and to more accurately define the β-selection checkpoint between DN3a and DN3b cells characterised by increased cell size and increased expression of CD27 and CD28, referencing additional relevant papers.

To better define the defect in WNK1-deficient thymocytes we have added a new Figure 2A in which we now separate DN3a and DN3b cells based on CD28 expression and cell size and show that loss of WNK1 does not affect the numbers of DN3a cells but causes a significant decrease in DN3b and DN4 cells, consistent with defective β-selection. However, we note that despite the reduced numbers of DN3b and DN4 cells, they are not completely absent. As shown in Figure 1C, DN4 cell numbers are reduced to ~40% of controls, and these residual cells have lost almost all *Wnk1* mRNA (Figure 1D). Thus, the loss of WNK1 is not equivalent to, for example, loss of RAG1 which results in a complete arrest of development at the DN3a stage just before the pre-TCR checkpoint. Instead, in the absence of WNK1, development can still proceed to the DN3b and DN4 stages, albeit with greatly reduced efficiency.

Figure 3B suggests that there is no impact of β-selection in fact since the DN3 to DN4 transition there seems unaffected.

Stimulation of WNK1-deficient *Rag1*^-/-^ thymocytes with anti-CD3 clearly induces early transcriptional and cell surface makers changes typical of transition of DN3a to DN4 cells: there is upregulation of *Cd27* and *Cd28* (new data added to Figure 4B) and downregulation of *Il2ra* (Figure 4B) accompanied by loss of CD25 from the surface (Figures 3B, 4C). Thus, the cells are phenotypically becoming DN4 cells. However, they do not expand in number and do not progress to the DP stage (Figure 2D), both of which are hallmark features of β-selection.

On the face of it, as the reviewer suggests, this looks different from the result in Figures 1B and 2A which shows a block between DN3a and DN3b, with reduced numbers of DN4 cells. However, the two experiments are different. In Figures 2D and 3B we analysed RAG1-deficient mice in which maturation from the DN3a compartment was acutely induced by injection of anti-CD3 antibody, resulting in a synchronous development of the cells into DN4 cells over a period of 48 h. In contrast, in Figures 1B and 2A the development across the β-selection checkpoint was driven by the normal physiological pre-TCR signal, and the analysis is of a thymus under steady state conditions, with continuous production of DN3a cells and then pre-TCR induced maturation of some of these into DN3b, DN4 etc. It is thus not surprising that there are some differences in the measured percentages of cells in the DN3 v DN4 compartments. Nonetheless, as noted above, even in the steady state conditions in Figures 1B and 2A some WNK1-deficient DN3a cells develop into DN3b and DN4 cells, but they are reduced in number. In the anti-CD3 induced development the cells become DN4 cells, but do not expand in number.

In our view, these two studies give a consistent picture that WNK1-deficient DN3a cells can differentiate into DN3b and DN4 cells, but that they then do not expand in number as control cells do, supporting our conclusion that WNK1 is required for the pre-TCR driven cell proliferation, but not the initial differentiation into DN3b and DN4 cells.

However, the DN4 subsets also need a clear and better definition since this is a mixture of cells if insufficient markers are included in the panel. This fraction may include γδ T cells and other lineages. This is also important for clearly understanding the frequency of TCRβ expressing thymocytes in DN3 and DN4 stages (Figure 2A).

We agree that it is very important to remove all the non-T cell lineage cells in order to get meaningful results on intracellular TCRβ expression in DN3 and DN4 cells. The DN subsets were defined as being positive for Thy1, but negative for a mixture of ‘Lineage’ antibodies specific for B220, CD3, CD4, CD8, CD11b, CD11c, CD19, DX5, GR-1, NK1.1, TCRβ and TCRγδ. Using this mixture, we excluded B cells, DCs, macrophages, NK cells, neutrophils as well as DP and SP thymocytes, and αβ and γδ T cells. The DN cells were then further separated by CD25 and CD44 for the DN1-4 subsets as shown in the manuscript. We note that DN4 cells were CD28+ as would be expected for bona fide DN4 thymocytes. Thus, we believe that the DN3 (a and b) and DN4 cells are well defined. This information is detailed in the Materials and methods.

Thus, it is extremely difficult to make clear conclusions on the precise developmental stage that is impacted by WNK1. This needs extensive revision in my opinion.2) Another major issue relates to the choice of the mouse model for conditional deletion of Wnk1. The authors use hCD2-iCre to mediate deletion of floxed Wnk1 gene and state that this deletion started at the DN2 stage. This is not in agreement with literature on this mouse model since hCD2-iCre expression has been detected in immature (ETP for T; pro-B for B lineage) and mature B/T cells, and also in bone marrow lymphoid progenitors (CLPs), natural killer cells and dendritic cells (Siegemund et al., 2015). In fact, the authors show in Figure 1D that significant reduction of Wnk1 mRNA expression was detected at DN1 stage of Wnk1fl/fl/hCD2-iCre thymocytes compared to the control. So, the authors need to consider the pleiotropic effects of conditional deletion of Wnk1 using hCD2-iCre mice which might be particularly important in relation to intrathymic DC development. Cpa3-Cre mice would allow T-specific deletion of Wnk1 (Feyerabend, Immunity. 2009). Alternatively, the authors could perform knockdown of Wnk1 in sorted wild type thymocytes and performed in-vitro co-culture assay to demonstrate the similar developmental defects observed in the mouse model.

We agree with the reviewer that CD2Cre is already active at the DN1 (ETP) stage in the thymus and we correct this point in the manuscript. Indeed, as noted by the reviewer, analysis of *Wnk1* mRNA in Figure 1D showed a 50% loss already in DN1 and DN2 cells. So, deletion is seen at DN1 stage, and may well occur earlier, but at these stages it is only partial. Deletion is not complete until DN3 stage (>90% loss of mRNA). We do not expect this 50% loss of *Wnk1* mRNA at the DN1 and DN2 stages to have an effect because heterozygous loss of *Wnk1* does not cause an arrest in thymic development.

Regarding the possibility that loss of *Wnk1* in DCs is affecting thymic development, this is possible in Figure 1A-C, but the experiment in Figure 1E-F (analysis of the *Wnk1*^D368A^ mutation) uses radiation chimeras made in sub-lethally irradiated RAG1-deficient mice, hence the DCs are largely *Wnk1*^+/+^ in these mice and unlikely to affect the phenotype.

3) There appear to be some inconsistencies in the data. In Figure 1A, the DN fraction of Wnk1-deleted thymi is much higher compared to the control (1.77% in Wnk1^fl/+^/hCD2-iCre vs. 55.8% in Wnk1^fl/fl^/hCD2-iCre). However, this is inconsistent with Figure 1B/C where none of the DN subsets have increased cell numbers or frequencies. In general, to understand the precise impact on cellularity and for clear interpretation of Figure 1, it is important to provide the data for total thymic cellularity.

As requested, we have added data on total thymic cellularity to Figure 1B,C. The *Wnk1* mutation causes a >90% reduction in the total number of thymocytes.

The DN fraction is increased as a percentage in the WNK1-deficient thymus in Figure 1A, but this is because there is huge loss of DP and SP thymocytes and greatly reduced total thymocytes numbers, not because there is an increase in the number of DN cells. Calculation of the numbers of DN cells shows no significant change in numbers of DN1 and DN2/3 cells, but a 60% reduction in the number of DN4 cells (Figure 1B, C). Note that in Figure 1C we show the *normalized* abundance of each subset in the WNK1-deficient thymus expressed as a percentage of the number of cells in the control thymi – this is not a graph of the % of cells in each subset. To make this clearer we have re-labelled the y-axis of the graph in Figure 1C as ‘Normalized cell number (% of control)’.

In Figure 2, the authors shown that DN4 cells are reduced in the KO. However, in Figure 3B, the authors show that there is no blockade or arrest in development of DN3 cells to DN4 stage. This is also inconsistent with the data in Figure 1B-C and may relate to the unclear definitions of the developmental stages.

As discussed above, the loss of WNK1 in the CD2Cre mice (Figure 1B-C) shows a 60% reduction in the number of DN4 cells. However, they are not all gone. In the experiments shown in Figures 2D and 3B development of RAG1-deficient DN3a cells is triggered by anti-CD3. This causes the DN3a cells to become DN4 cells (Figure 3B), but there is no expansion of cell numbers or differentiation into DP cells (Figure 2D). Overall the data in the two studies give a consistent picture that WNK1 is required for pre-TCR driven cell proliferation, but not the initial differentiation into DN3b and DN4 cells.

4) It is not clear to me how many mice were analyzed and how the data is presented. For instance, the legend of Figure 1 states that data are from one of two experiments but according to the reporting form this should be more given that 5-7 mice were analyzed according to that same legend.

Each figure legend lists the numbers of mice being analysed in each part of the figure and explains whether this was data from one experiment out of several, or whether the data has been pooled from more than one experiment. Individual experiments contained multiple animals. We have edited the legends to make this clearer. For example, the graphs in Figure 1B-D, F show data pooled from two independent experiments, as stated in the legend.

5) In general, there is insufficient direct evidence for the proposed regulatory axis as proposed by the authors. The data mainly shows indirect relationships and the paper would strongly benefit from functional rescue experiments to support these strong conclusions.

We have added a new experiment in Figure 9, addressing the question of whether ectopic expression of MYC can rescue the developmental block in WNK1-deficient thymocytes.

We generated bone marrow radiation chimeras using bone marrow from mice with a floxed allele of *Wnk1* and tamoxifen-inducible Cre and infected these cells with retroviral vectors expressing MYC, MYC-T58A (a stabilised form of MYC) or with an empty vector. Six weeks after reconstitution we induced deletion of *Wnk1* with tamoxifen and analysed thymi 7 days later. Analysis shows that once again we see that loss of WNK1 results in a drop in the number of DN3b, DN4, ISP and DP cells (Figure 9D).

The efficiency of retroviral transduction in the chimeras (as detected by % of GFP^+^ cells) varied significantly from mouse to mouse. Consequently, analysis of cell numbers following expression of MYC or MYC-T58A would not generate meaningful data to assess whether MYC rescues WNK1-deficient thymocytes. However, by normalizing the % GFP^+^ cells in DN3b and DN4 subsets to the % GFP^+^ cells in the DN3a subset in each individual mouse, we were able to evaluate if expression of MYC or MYC-T58A provides a selective advantage to cells transiting through the pre-TCR checkpoint.

Importantly, we found that control or WNK1-deficient thymocytes with the empty vector became less abundant as cells developed from DN3a to DN3b and to DN4. In contrast, cells infected with MYC or MYC-T58A-expressing retroviruses became more abundant (Figure 9E, F). Collectively, this new data demonstrates that ectopic expression of MYC or MYCT58A confers a selective advantage to both control and WNK1-deficient thymocytes developing across the pre-TCR checkpoint, which is consistent with our hypothesis that amounts of MYC are limiting in the absence of WNK1.

Reviewer #2:[…] (1) The authors clearly establish that MYC protein levels in thymocytes are lower in the absence of WNK1, however it would be important to show whether ectopic expression of MYC could lead the rescue of the proliferation and/or differentiation defect. This can be easily performed, as per Wong et al. (PMID: 22649105), a related paper that examined the role of pre-TCR, Notch and MYC during T cell development.

To address this issue, we adopted an in vivo approach to test the hypothesis that MYC can rescue the development of WNK1-deficient thymocytes across the pre-TCR checkpoint. This new data is included in Figure 9. For a detailed response to this comment, please see reviewer 1, major comments point 5.

2) The clear effect on pre-TCR signaling and the block in the DN to DP differentiation begs the question as to whether γ/δ-T cell differentiation is similarly affected or not, which would position WNK1 as uniquely required for α/β-T lineage differentiation or as shared signaling axis for both lineages. Simply showing whether γδ T cells are present in the thymus and peripheral tissues would directly address this point.

We have added new data on the numbers of γδ thymocytes to Figure 1B, C. The loss of WNK1 does not affect their numbers.

3) It curious that the authors did not directly address whether the absence of WNK1 in DN cells has an effect on any aspects of ion transport.

This would be a very interesting parameter to measure, however this has not been possible for technical reasons. There are fluorescent dyes that are claimed to report concentrations of Na^+^ and K^+^ ions, but we have not been able to get them to work. We have had some success in measuring the amounts of Na^+^ and K^+^ in T cell lines using inductively coupled plasma mass spectrometry, but we have not yet managed to get this to work in mature T cells and this is not currently feasible with the very small numbers of DN thymocytes.

Reviewer #3:[…] This study is well designed, executed, and thoughtfully interpreted and brings new understanding to signaling axes controlling T cell development. There are however, a few issues that would strengthen the manuscript if addressed.1) The important role played by STK39 is somewhat remarkable given how little appears to be expressed in early thymocytes. Could the authors please comment on this?

We too were surprised by this. Based on the RNAseq we expected that OXSR1 would be the dominant kinase and that mutation of STK39 would have little or no effect. One possibility is that despite low levels of *Stk39* mRNA, the amount of STK39 protein may be closer in amount to OXSR1 because of post-transcriptional effects (e.g. translation, protein stability). This could be looked at by immunoblotting, but we have not been able to identify good antibodies to STK39.

2) It would be useful for the authors to measure by flow the levels of MYC in the SLC12A2-ko thymocytes.

We agree with the reviewer that this would have been a good experiment to add. Unfortunately, this has not been possible due to the Covid-19 pandemic – we had to remove the Slc12a2 mutant mice from the breeding unit.

3) While the role of MYC in controlling cell size, cell proliferation, and thymocyte development is well documented, it would be a nice finishing touch to test whether ectopic expression of MYC would rescue the developmental defects seen in the Wnk1-deficient thymocytes, if only in vitro in OP9 cultures.

To address this issue, we adopted an in vivo approach to test the hypothesis that MYC can rescue the development of WNK1-deficient thymocytes across the pre-TCR checkpoint. This new data is included in Figure 9. For a detailed response to this comment, please see reviewer 1, major comments point 5.